https://doi.org/10.1038/s41467-019-12064-1　　**OPEN**

# Typhoid toxin exhausts the RPA response to DNA replication stress driving senescence and *Salmonella* infection

Angela E.M. Ibler[1,2], Mohamed ElGhazaly[1], Kathryn L. Naylor[1], Natalia A. Bulgakova [1], Sherif F. El-Khamisy [3,4] & Daniel Humphreys [1]

*Salmonella* Typhi activates the host DNA damage response through the typhoid toxin, facilitating typhoid symptoms and chronic infections. Here we reveal a non-canonical DNA damage response, which we call RING (response induced by a genotoxin), characterized by accumulation of phosphorylated histone H2AX (γH2AX) at the nuclear periphery. RING is the result of persistent DNA damage mediated by toxin nuclease activity and is characterized by hyperphosphorylation of RPA, a sensor of single-stranded DNA (ssDNA) and DNA replication stress. The toxin overloads the RPA pathway with ssDNA substrate, causing RPA exhaustion and senescence. Senescence is also induced by canonical γH2AX foci revealing distinct mechanisms. Senescence is transmitted to non-intoxicated bystander cells by an unidentified senescence-associated secreted factor that enhances *Salmonella* infections. Thus, our work uncovers a mechanism by which genotoxic *Salmonella* exhausts the RPA response by inducing ssDNA formation, driving host cell senescence and facilitating infection.

---

[1] Department of Biomedical Science, University of Sheffield, Sheffield S10 2TN, UK. [2] Department of Pathology, Tennis Court Road, University of Cambridge, Cambridge CB2 1QP, UK. [3] The Healthy Life Span Institute, Department of Molecular Biology and Biotechnology, University of Sheffield, Sheffield S10 2TN, UK. [4] Center of Genomics, Zewail City of Science and Technology, Giza, Egypt. Correspondence and requests for materials should be addressed to D.H. (email: d.humphreys@sheffield.ac.uk)

D amage to our genomes, arising endogenously during DNA replication, or exogenously from genotoxins, often generates DNA breaks causing mutations and chromosomal aberrations, which underlie diseases[1]. To counteract DNA damage and its transmission during cell division, cells detect DNA damage through the DNA damage response (DDR) that orchestrates repair and cell fate decisions including survival, apoptosis and senescence[1–4].

Master regulators of the DDR are the kinases ATM (ataxia-telangiectasia mutated) and ATR (ATM and rad3-related) that phosphorylate diverse substrates to regulate DNA repair and the cell-cycle[1]. In a canonical DDR, ATM responds to DNA double strand breaks (DSBs) while ATR senses stress exerted during DNA replication but interconnected functions are described[5]. Replication stress is predominantly caused by damaged DNA replication forks, an aberrant structure yielding single-stranded DNA (ssDNA) that recruits the ssDNA-binding protein complex RPA[6] (replication protein A). RPA coats and protects ssDNA while recruiting and activating ATR[7] to further counteract DNA damage arising from replication stress[8]. The ATR pathway pauses the cell cycle, halts firing of DNA replication at new sites and stabilises forks for repair. This is key as persistent replication stress converts forks into DSBs, the most lethal form of damage that is counteracted by the ATM pathway[1,5]. Uniting the functions of ATM and ATR is phosphorylation of their effector, histone H2AX at S139 ($\gamma$H2AX)[9,10], which represents a signature of DDRs[1,5]. $\gamma$H2AX recruits and retains DDR proteins at sites of DNA damage to coordinate repair and cell fate[1–4].

Given the importance of the DDR in coordinating diverse cellular processes, it is perhaps not surprising that pathogenic bacteria have evolved sophisticated ways to manipulate the host DDR to execute virulence strategies and establish infections[11]. This is exemplified by genotoxic strains of Salmonella[12–16], particularly the human-adapted intracellular pathogen Salmonella Typhi that underlies 21 million infections and 200,000 deaths each year[17]. The problem is exacerbated by chronically infected carriers, like the infamous typhoid Mary, who do not exhibit typhoid symptoms but retain S.Typhi in the population and transmit the pathogen through contaminated faeces[17]. The mechanisms by which the two disease manifestations, typhoid fever and chronic carriage, are mediated remain unclear. Experiments in animal models showed that the mortal symptoms of typhoid and chronic carriage were facilitated through a virulence factor called the typhoid toxin[13,14]. The toxin enters human cells where it is thought to cause DNA damage through putative nuclease activity[12,18].

The typhoid toxin is unique, comprising a pentameric PltB subunit linked to PltA, which is bound to the genotoxic DNase1-like subunit CdtB[12,13], a subunit found in related cytolethal distending toxins (CDTs). Canonical CDTs possess DNA nickase activity implicated in the direct formation of DSBs and ssDNA breaks (SSBs)[19–21] that activate ATM/ATR pathways and induce G1 and G2 cell-cycle arrest[22–25]. Toxigenic Salmonellae invade human host cells where they reside within Salmonella-containing vacuoles and express the typhoid toxin that is exocytosed into the extracellular milieu[12,26,27]. Once deployed, the PltB subunit binds bystander cell surface receptors to trigger uptake after which the CdtB traffics to the nucleus and induces DDRs marked by $\gamma$H2AX, cell cycle arrest and cellular distension[12,18,28]. The typhoid toxin has been shown to induce fever and mortality in animal models[13], and facilitate systemic infections and chronic Salmonella carriage[14,16]. Understanding the toxin is especially important as it is also encoded by related Salmonella serovars including S. Paratyphi and S. Javiana that cause disease in humans and food-chain animals worldwide[15,29].

Despite its importance, the cellular DDR responses to the typhoid toxin are unclear and the mechanisms by which it promotes infection are not understood. Here, we report a non-canonical DDR to ssDNA induction by the typhoid toxin that drives senescence and infection.

## Results

**A non-canonical DDR induced by the typhoid toxin.** To study host cell responses to the typhoid toxin, we purified recombinant holotoxin (henceforth toxin) from Escherichia coli and confirmed the presence of epitope-tagged PltB$^{HIS}$, PltA$^{Myc}$, and CdtB$^{FLAG}$ by SDS-PAGE and immunoblotting (Fig. 1a, b, Supplementary Fig. 1A, B). The typhoid toxin is known to trigger cell cycle arrest and the DDR in mammalian cells[12,18,28]. Thus, asynchronous human HT1080 cells were intoxicated for 2 h followed by a chase period of 24 h. Cell cycle progression was examined using the DNA-binding dye propidium iodide and flow cytometry to determine DNA content (Fig. 1c, Supplementary Fig. 1C, D). Untreated cells progressing through the cell cycle were predominantly observed in G1 (~75%) with ~13% in S-phase and ~17% in G2. In contrast, intoxication with wild-type toxin (toxin$^{WT}$) triggered cell-cycle arrest (Fig. 1c: G1, ~30%; G2/M, ~60%; S ~10%, Supplementary Fig. 1E). This response was driven by the catalytic activity of CdtB as mutation of H160 (toxin$^{HQ}$) had no effect on cell cycle progression, which was equivalent to control (Fig. 1c, Supplementary Fig. 1E).

DDR activation is marked by phosphorylation of H2AX at S139 ($\gamma$H2AX)[9,10]. Immunofluorescence microscopy showed that toxin$^{WT}$ triggered formation of $\gamma$H2AX, which accumulated in distinct foci (Fig. 1d: white arrows). This was expected as a classical DNA damage checkpoint has been previously observed for the typhoid toxin and related CDTs[12,18,25,28,30,31]. To our surprise, we found that $\gamma$H2AX was not observed in foci in a significant proportion of intoxicated host cells (Fig. 1d). Instead, $\gamma$H2AX was found enriched in rings circling the edge of the nuclear periphery (blue arrows), which was also observed at high resolution by Airyscan confocal microscopy (Fig. 1e). Both $\gamma$H2AX phenotypes were mutually exclusive as foci were never observed in cells with rings (exemplified by the arrows in Fig. 1d). Rings of $\gamma$H2AX were neither observed in control cells nor in cells treated with toxin$^{HQ}$ (Fig. 1d).

To examine the $\gamma$H2AX phenotype more closely, we established an image-based 'RING Tracking' method, which automated segmentation of $\gamma$H2AX phenotypes and quantified $\gamma$H2AX rings in cell populations (Supplementary Fig. 2). Classical $\gamma$H2AX foci were present in ~15% of intoxicated cells while non-canonical $\gamma$H2AX rings were found in ~50% (Fig. 1f). Rings were observed in all cell lines tested, varying between ~25% and ~55% of cells (Supplementary Fig. 3A). This included mouse embryonic fibroblasts (MEFs) imaged by high-resolution structured illumination microscopy (Supplementary Fig. 3B). Increasing the dose 10-fold from 5–50 ng/ml had no additive effect on $\gamma$H2AX rings (Fig. 1g), which was maximal in 50% of HT1080s suggesting an unidentified limiting factor. In contrast, decreasing the toxin dose 10-fold to 0.5 ng/ml resulted in loss of $\gamma$H2AX rings, which could be significantly increased by extending the toxin$^{WT}$ incubation from 2–8 h. Thus, rings of $\gamma$H2AX are induced in a dose-dependent manner.

$\gamma$H2AX localisation at the nuclear periphery has been reported as a caspase-dependent marker of early apoptosis[32,33]. However, in contrast to the known apoptosis inducer staurosporine, neither toxin$^{WT}$ nor toxin$^{HQ}$ induced significant apoptotic cell death, activation of the caspase cascade or cleavage of the caspase substrate PARP1 (Supplementary Fig. 4A, B). Moreover, while the pan-caspase inhibitor ZVAD-OMe-FMK blocked caspase

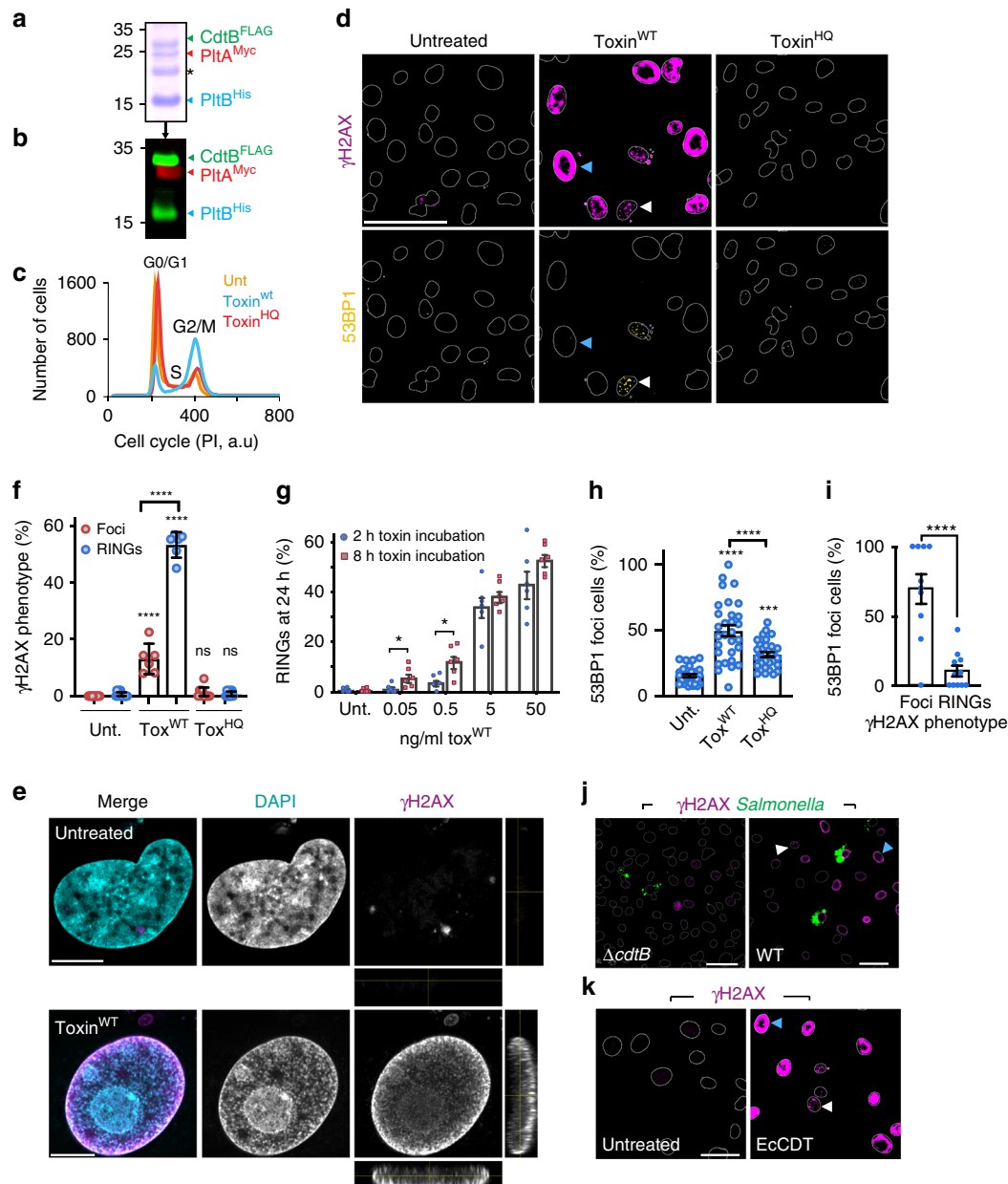

**Fig. 1** Non-canonical DNA damage response induced by a bacterial genotoxin. **a** Recombinant typhoid toxin comprising epitope-tagged CdtB$^{FLAG}$, PltA$^{Myc}$ and PltB$^{His}$ (asterisk marks *E.coli* contaminant). MW in kDa left. **b** Toxin immunoblotted with anti-FLAG (CdtB$^{FLAG}$, green), -Myc (PltA$^{Myc}$, red), or -His antibodies (PltB$^{His}$, green). MW in kDa left. HT1080s treated with 5 ng/ml wild-type toxin (toxin$^{WT}$) or catalytically inactive toxin$^{HQ}$ for 2 h before analysis at 24 h as follows: **c** Representative cell-cycle arrest induced by toxin$^{WT}$. Propidium iodide (PI) used as DNA marker and fluorochrome (a.u, arbitrary units). Cell-cycle phases indicated. **d** Representative image of γH2AX (magenta) and 53BP1 (yellow) with outlines of DAPI-stained nuclei. Scale bars 50 μm. **e** Representative confocal microscopy of γH2AX RING (magenta) and DAPI-stained nuclei (cyan). Side-panels show z-sections. Scale bars 5 μm. **f** Proportion of cells with γH2AX RINGs and foci. Coloured circles indicate means from technical replicates (2 biological replicates, ~2000 nuclei/variable). Error bars Standard Deviation (SD). **g** Toxin$^{WT}$-induced γH2AX RINGs. Incubation times and concentrations indicated. Six technical replicates, ~360 nuclei/variable (one biological replicate). Error bars Standard Error of the Mean (SEM). **h** Proportion of cells with 53BP1 foci. Coloured circles indicate means from fields of view (3 biological replicates, ~2200 nuclei/variable). Error bars SEM. **i** Proportion of 53BP1-positive cells with γH2AX RINGs and foci. Coloured circles indicate means from technical replicates (two biological replicates, 450 nuclei/variable). Error bars SEM. **j** Representative image of γH2AX RINGs (magenta) in infected cells. *Salmonella* Typhi (green) encoding the toxin (WT) or toxin null mutant (Δ*cdtB*) at 24 h shown with outlines of DAPI-stained nuclei. Scale bars 50 μm. **k** Representative image of γH2AX RINGs (magenta) in EcCDT-treated cells shown with outlines of DAPI-stained nuclei. Scale bars 25 μm. In representative images **d, j, k** arrows indicate DNA damage RINGs (blue) and foci (white). Statistical significance (**** = $P < 0.0001$, *** = $P < 0.0002$, ** = $P < 0.0021$, * = $P < 0.0332$, ns denotes non-significant = $P >= 0.05$) calculated using an unpaired two-sided *t* test (**g, i**), or relative to corresponding control using one-way ANOVA and Tukey's multiple comparison test (**f, h**). Source data provided as a Source Data file

activation (Supplementary Fig. 4C), it had no effect on toxin-induced γH2AX rings (Supplementary Fig. 4D) confirming their independence from apoptosis.

γH2AX acts as a hub by recruiting DNA repair proteins such as 53BP1[1–3] (p53-binding protein 1). Indeed, 53BP1 was either diffusely localised across the nucleus or below detection in untreated controls (Fig. 1d, Supplementary Fig. 5A) but in intoxicated cells (toxin[WT]), the proportion of cells with 53BP1 foci significantly increased (Fig. 1d, h). Relative to untreated controls, toxin[HQ] induced a modest but reproducible increase in γH2AX (Fig. 1d) and 53BP1 foci (Fig. 1d, h), which was likely due to LPS contamination that binds His-tagged proteins and activates DDRs[34–36]. The γH2AX foci in toxin[WT]-treated host cells invariably co-localised with 53BP1 (white arrows) but not in untreated controls (Fig. 1d). In toxin[WT]-treated cells displaying γH2AX rings however, no 53BP1 co-localisation was apparent (blue arrows) showing that 53BP1 had uncoupled from the γH2AX pathway (Fig. 1d, i). The uncoupling of γH2AX from the canonical 53BP1 pathway and its localisation in rings at the nuclear periphery mark the discovery of a non-canonical host cell γH2AX **r**esponse **in**duced by a **g**enotoxin (henceforth RING).

**RINGs are triggered by Salmonella and related genotoxins.** We next examined γH2AX RINGs during bacterial infection. *Salmonella* enterica serovars Typhi, Paratyphi, Javiana and Montevideo are known to target host cells with the typhoid toxin[12,28,37]. Intracellular toxigenic *Salmonella* secrete the typhoid toxin into the extracellular environment from where the toxin intoxicates bystander host cells[12,16]. When we infected HT1080 cells with toxigenic strains of *S*.Typhi or *S*.Javiana, DDRs were observed as γH2AX foci at 24 h post infection (white arrows in Fig. 1j). Remarkably, infection also induced RINGs (blue arrows in Fig. 1j), which increased from ~25% at 24 h to ~50% by 48 h and were not induced by toxin-deficient strains (Δ*cdtB*) (Fig. 1j, Supplementary Fig. 5B, C).

The typhoid toxin is a unique chimera of CdtB, the toxigenic subunit of CDTs, and pertussis toxin[12,13]. CDTs comprise CdtA, CdtC and CdtB that are encoded by diverse bacterial pathogens[25]. Remarkably, like the typhoid toxin, purified CDT from pathogenic *E.coli* (EcCDT) also induced both DNA damage foci (white arrow), as previously reported[25], and RINGs in ~30% of intoxicated host cells at 24 h (blue arrows) (Fig. 1k). Thus, RING formation is a phenomenon conserved amongst canonical and non-canonical CDTs.

**RINGs are triggered in S-phase of the cell cycle.** We next addressed the mechanisms by which γH2AX RINGs develop. 53BP1 localises to pre-replicative chromatin and is negatively regulated in S-phase[38], which may explain the exclusion of 53BP1 from RINGs (Fig. 1d, i). Our interest in S-phase was furthered by the specific localisation of γH2AX in RINGs that was reminiscent of heterochromatin, which is refractory to γH2AX production until late S-phase when heterochromatin is decompacted for replication[1,39,40]. Indeed, γH2AX in RINGs co-localised with heterochromatin marked by histone-3 lysine-9 methylation (Supplementary Fig. 5D, E: H3K9me3). Thus, we next examined the significance of S-phase in RING formation.

We arrested cells in G1 by serum starvation to block entry into S-phase (Supplementary Fig. 6A) before assaying RING formation (Fig. 2). In the presence of serum (+serum), RINGs were observed in ~20% of cells at 6 h and ~50% by 48 h (Fig. 2a). When cells were prevented entry into S-phase (− serum), RING formation was completely abolished and the DNA damage phenotype was limited to γH2AX foci (Fig. 2a, b). When intoxicated serum-starved cells were permitted entry into S-phase

by addition of serum at 24 h (released), we observed RINGs in ~40% of cells by 48 h (Fig. 2a, c, d) showing a requirement for S-phase in RING formation at heterochromatin.

To address whether RINGs were generated in S-phase, we used fluorescently labelled nucleotide analogues BrdU and EdU to assay DNA synthesis during RING formation. Toxin[WT] inhibited global BrdU incorporation confirming a reduction in DNA synthesis (Supplementary Fig. 6B, C). Two hours intoxication before 24 h incubation with EdU demonstrated DNA replication in RING cells by 24 h (Fig. 2e, f). Interestingly, γH2AX foci cells contained no EdU demonstrating G1 arrest (Fig. 2e) that were observed by flow cytometry (Fig. 1c, Supplementary Fig. 1E). This further confirms that γH2AX/53BP1 foci in Fig. 1d mark 53BP1-labelled damage on pre-replicative chromatin[38], which peaked in serum-starved cells (Supplementary Fig. 6D). RINGs were maximal at 48 h (Fig. 2a, Supplementary Fig. 5B), thus, we reasoned that a short 1 h pulse with EdU at 24 h when RINGs were reaching maximum would also reveal damage during S-phase, which was likely still in progress in a sub-population of cells. Indeed, a significant population of RING cells with newly synthesised DNA were observed (Fig. 2f: 1 h) establishing that RINGs signify damage in S-phase.

**RINGs are dependent upon regulator of replication stress ATR.** We hypothesised that RINGs signal activation of ATR that counteracts DNA replication stress by phosphorylating diverse substrates including H2AX and CHK1[1,5]. To resolve whether ATR determines the specific localisation of γH2AX at RINGs, we intoxicated cells in the presence of ATR (iATR) and ATM (iATM) inhibitors (Fig. 2g, h, Supplementary Fig 6E, F). Remarkably, RINGs were driven through ATR (Fig. 2g: blue arrows, 2H), while γH2AX foci that mark G1-arrested cells required ATM (Fig. 2g: white arrows, 2H). Moreover, activated ATR and ATM were observed in intoxicated cells (Fig. 3a, Supplementary Fig. 6E, F). Consistent with ATR-driven RING formation, phosphorylation of CHK1 (pCHK1) and γH2AX were induced in RING permissive conditions (Fig. 2d: +serum, Supplementary Fig. 6G). In the presence of serum, bulk analysis by immunoblotting consistently revealed maximal γH2AX signal (Fig. 2d: +serum), which was consistent with fluorescent images (Fig. 2b). This likely reflects the presence of both γH2AX foci and RINGs in populations with serum whereas serum-starved cells only have foci and thus show less γH2AX signal (Fig. 2b, d). Indeed, both CHK1 activation and γH2AX signal were inhibited by serum-starvation and restored in released cells (Fig. 2d). Similarly, both RINGs and pCHK1/γH2AX were apparent in replicating THP1 monocyte cells (-PMA) but were absent in terminally differentiated non-replicating THP1 macrophage cells (+PMA) (Supplementary Fig. 6H, I). Taken together, these findings show that the typhoid toxin triggers DNA damage foci in G1 through ATM but cell entry into S-phase induces ATR-dependent RING formation in response to replication stress.

**ssDNA sensor RPA accumulates on the chromatin of RING cells.** The heterotrimeric complex RPA (Replication Protein A70, 32 and 14) senses replication stress by coating ssDNA at damaged replication forks where it activates ATR[5,7]. If replication stress is not resolved then damaged forks can collapse leading to DSBs. Damaged replication forks are marked by ATR-mediated hyper-phosphorylation of ssDNA-bound RPA32 on multiple sites including T21 and S33[41,42], which facilitate recruitment of repair factors to stalled forks[43].

To study the significance of RPA in RING formation, we examined phosphorylation of ssDNA-bound RPA in intoxicated cells (Fig. 3a, Supplementary Fig. 7). Consistent with ATR

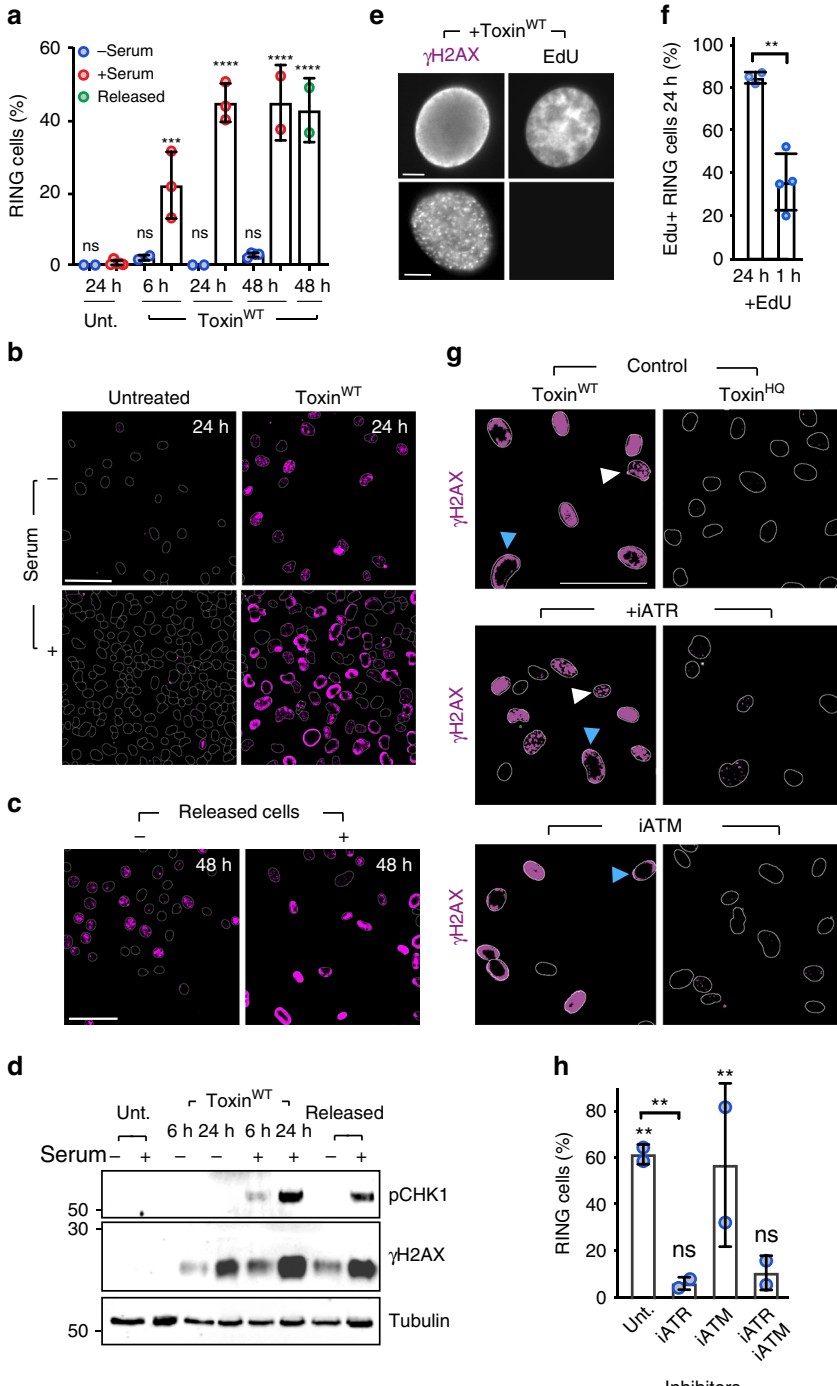

**Fig. 2** RINGs are triggered during S-phase of the cell cycle. **a** Proportion of toxin-induced γH2AX RINGs in non replicating (− serum, blue circle) or replicating (+ serum, red circle) HT1080 cells. Untreated control cells are shown. Serum-starved cells were incubated with serum from 24 h as indicated (released, green circle). Coloured circles indicate means from at least two biological replicates (2000 nuclei per variable). **b** Representative image of γH2AX RINGs ± serum at 24 h. γH2AX labelled in magenta. Outlines of DAPI-stained nuclei shown. Scale bars 50 μm. **c** Representative image of γH2AX RINGs in released cells. Labelled as B. Scale bars 50 μm. **d** Representative immunoblot of toxin-induced phosphorylation of CHK1 (pCHK1) and H2AX (γH2AX) ± serum at 6 or 24, or 48 h (released). MW in kDa left. **e** Representative image of EdU-labelled DNA in γH2AX RING or foci cells at 24 h. EdU incubated for 24 h. Scale bars 5 μm. **f** Proportion of γH2AX RING cells with EdU-labelled DNA at 24 h. EdU incubated for 24 h or 1 h before fixation. Circles indicate means from at least three biological replicates (>220 nuclei per condition). **g** Representative image of γH2AX RINGs (blue arrow) or foci (white arrow) in the presence of DMSO (control), ATR inhibitor (iATR), or ATM inhibitor (iATM) shown with outlines of DAPI-stained nuclei. Toxin[HQ] used as negative control. Scale bars 50 μm. **h** Proportion of γH2AX RINGs in presence of ATR/ATM inhibitors at 24 h. Control toxin[HQ]-treated cells are shown. Circles indicate means from two biological replicates (470 nuclei per variable). Statistical significance (**** = $P < 0.0001$, *** = $P < 0.0002$, ** = $P < 0.0021$, * = $P < 0.0332$, ns = $P > = 0.05$) was calculated relative to corresponding control (unt, +serum in **a**; toxin[HQ] in **h**) using one-way ANOVA and a Tukey's multiple comparison test (**a, h**), or between samples using an unpaired two-sided *t* test (**f**). Error bars SD. Source data are provided as a Source Data file

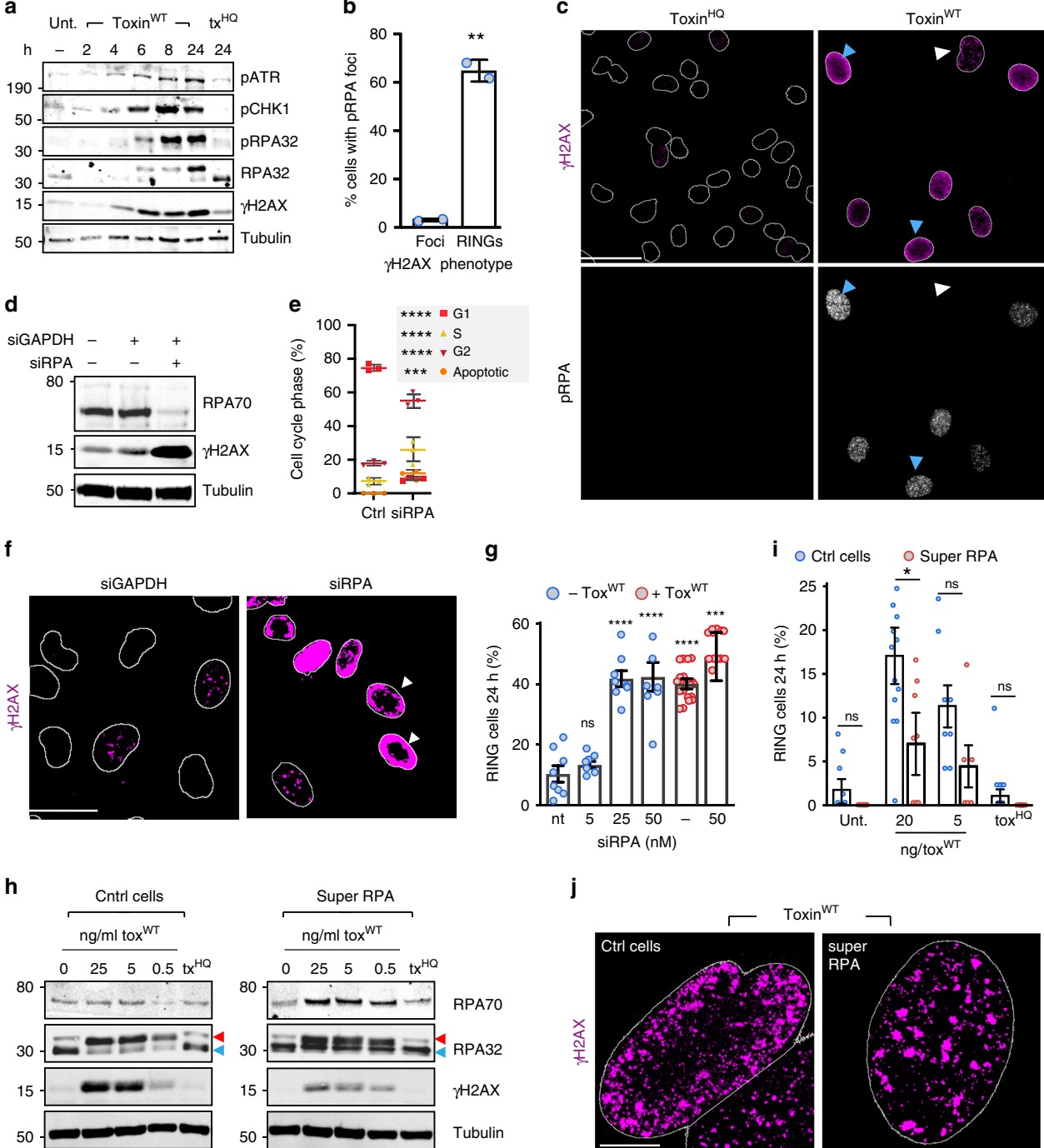

**Fig. 3** RINGs mark RPA exhaustion induced by the typhoid toxin. **a** Representative immunoblot of HT1080 cells intoxicated with toxin[WT] or toxin[HQ] (tx[HQ]) for 2 h before analysis with indicated antibodies at time-points marked in hours (hr). MW in kDa left. **b** Proportion of intoxicated cells with γH2AX RINGs or γH2AX foci containing RPA pT21 foci (pRPA, grayscale). γH2AX labelled in magenta. 2 biological replicates (1500 nuclei per variable). Error bars SD. **c** Representative image of RPA32 pT21 (pRPA) with γH2AX RINGs (blue arrow) or foci (white arrow). Scale bars 50 μm. **d** Representative immunoblot of RPA knockdown (siRPA) cells shown with transfection (siGAPDH) and gel loading (tubulin) controls. In all, 25 nM siRNA for 48 h. MW in kDa left. **e** Cell-cycle phases in RPA knockdown or siGAPDH control (ctrl) cells. Coloured symbols indicate means from 3 biological replicate (60,000 cells per variable). Error bars SD. **f** Representative image of γH2AX RINGs (white arrow) in RPA knockdown cells. Scale bars 25 μm. **g** Proportion of γH2AX RINGs in RPA knockdown cells ± toxin[WT]. Non-transfected control (nt). Circles indicate means from fields of view in two biological replicates (122 nuclei per variable). Error bars SEM. **h** Representative immunoblot of super-RPA cells ± toxin[WT] analysed with indicated antibodies. Hyper-phosphorylated RPA32 (red arrow), non-phosphorylated RPA32 (blue arrow). MW in kDa left. **i** Proportion of γH2AX RINGs in super-RPA cells ± toxin[WT]. Two biological replicates (2000 nuclei per variable). Error bars SEM. **j** Representative image of γH2AX in U2OS control cells (ctrl) or super RPA U2OS cells stably expressing recombinant RPA complex intoxicated with 5 ng/ml toxin[WT]. Scale bars 5 μm. In representative images **c**, **f**, **g** outlines of DAPI-stained nuclei are shown. Statistical significance (**** = $P < 0.0001$, *** = $P < 0.0002$, ** = $P < 0.0021$, * = $P < 0.0332$, ns = $P \geq 0.05$) was calculated between indicated experiments using an unpaired two-sided $t$ test (**b**), or calculated relative to indicated controls using one-way ANOVA (**g**, **i**) or two-way ANOVA (**e**) together with a Tukey's multiple comparison test. Source data are provided as a Source Data file

activation at damaged forks, RING formation was marked by hyper-phosphorylation of RPA32 that migrated more slowly than the RPA32 in untreated or toxin[HQ]-treated cells. Indeed, only toxin[WT] triggered phosphorylation of S33 (pRPA32). RPA phosphorylation was first observed at 6 h and continued to accumulate over 24 h, which was also found for ATR activation and its targets CHK1 and H2AX (Fig. 3a). Finally, when we examined RPA loading onto chromatin by cell extraction and fluorescence microscopy, RPA32 pT21 (pRPA) foci were observed confirming that RING-positive cells contained damaged replication forks (toxin[WT]), which were not found in the control (toxin[HQ]) (Fig. 3b, c). Strikingly, RPA foci were restricted to those with RINGs (blue arrows; ~ 65% pRPA positive) and not γH2AX foci (white arrows; ~3% pRPA positive) (Fig. 3b, c).

**Typhoid toxin triggers RINGs by causing RPA exhaustion.** RPA protects ssDNA from breakage[5,42,44] and is phosphorylated in response to accumulating ssDNA[5,7]. The typhoid toxin possesses endonuclease activity (Supplementary Fig. 8A) which, in related CDTs, is known to introduce ssDNA breaks[19,21]. Physical DNA breaks were also observed in intoxicated cells (Supplementary Fig. 8B, C), which is consistent with RING formation by EcCDT (Fig. 1K) that is known to mimic DNA damaging agents by causing DNA breaks[23]. Thus, phosphorylated RPA likely signified accumulating ssDNA in response to toxin nuclease activity. Consequently, we predicted that depleting the pool of RPA by siRNA transfection (siRPA) would sensitise cells to the typhoid toxin and RING formation (Fig. 3d–g, Supplementary Fig. 8D). To our surprise, RPA knockdown alone was sufficient to cause RINGs (Fig. 3f: white arrow), which was co-incident with increased γH2AX (Fig. 3d) and G2 cell-cycle arrest (Fig. 3e) establishing that siRPA phenocopied the typhoid toxin (Fig. 3g). Like toxin-induced RING formation (Fig. 2g, h), ATR was critical to siRPA-induced RINGs, which were impeded by siATR transfection (Supplementary Fig. 8E).

RPA binds ssDNA in stoichiometric fashion and is considered rate limiting for protecting ssDNA at replication forks[5,42]. Thus, we hypothesised that insufficient RPA available to bind ssDNA was the common denominator for RING-induction, which was driven by loss of RPA itself (i.e. siRPA) or by unregulated supply of ssDNA (i.e. via toxin nuclease activity). Indeed, when RPA-knockdown cells were treated with toxin, there was no additive effect in RINGs (Fig. 3g: red circles). Moreover, low concentrations of siRPA (Fig. 3g: 5 nM) and tox[WT] (Fig. 1f: 0.05 ng/ml) that were normally insufficient for RING formation, synergised to induce RINGs (Supplementary Fig. 8F) further establishing that RINGs are formed via RPA exhaustion.

We hypothesised that increasing the cellular pool of RPA would defend cells against the toxin and RING formation. To test this possibility, we intoxicated U2OS cells engineered to stably express additional RPA complex[44], which we confirmed by immunoblotting (Fig. 3h, Supplementary Fig. 9) before imaging and RING Tracking (Fig. 3i, j). Intoxicated U2OS control cells (ctrl) formed RINGs but by extending the dynamic range of ssDNA protection in 'super RPA cells', RING formation was reduced by ~50% (Fig. 3i), which is exemplified in Fig. 3j. No RINGs were observed with toxin[HQ]. A similar trend was also observed in bulk analysis by immunoblotting where toxin-induction of γH2AX was markedly reduced in super RPA cells, approximately ~50% (Fig. 3h, Supplementary Fig. 9D), which mirrored the reduction in RINGs (Fig. 3i). This was likely due to a delay in RPA exhaustion as indicated by the abundance of non-phosphorylated RPA32 (blue arrow) in the population of super RPA cells relative to toxin[WT]-treated control cells where RPA was predominantly hyper-phosphorylated (red arrow) (Fig. 3h).

Indeed, the enzymatic activity of toxin[WT] was still able to overcome super RPA cells, which generated RINGs and failed to resume proliferation following intoxication (Fig. 3i, Supplementary Fig. 10A, B). Together, these findings show that RING formation is a consequence of RPA exhaustion.

**Typhoid toxin exhausts RPA by oversupply of ssDNA substrate.** The findings support the view that RINGs were induced by an imbalance between RPA and ssDNA substrate, which caused RPA exhaustion. To study RPA exhaustion more directly, we developed an assay that exploited ssDNA protection by RPA (depicted in Supplementary Fig. 10C). In the RPA protection assay, we predicted that exposed ssDNA would act as a DNA template for polymerisation of BrdU-labelled DNA by exogenous DNA polymerase, which would be sterically hindered by coats of RPA loaded on ssDNA. Thus, DNA polymerase would preferentially access unprotected tracks of ssDNA in RPA-exhausted cells. The toxin[WT] induced significantly more ssDNA-derived BrdU foci (ssDNA[BrdU]) than control toxin[HQ] (Fig. 4a, b). Aphidicolin (APH), an inhibitor of DNA replication that generates extended tracks of ssDNA by inhibition of DNA polymerases[45], also induced ssDNA[BrdU] (Fig. 4b, Supplementary Fig. 10D) supporting the view that the toxin generates ssDNA. Moreover, RPA foci in toxin[HQ]-treated cells lacked ssDNA[BrdU] indicating RPA protection of ssDNA (Fig. 4a, magnified inset) while toxin[WT] generated ssDNA[BrdU] foci that were either partially protected (white arrow indicating RPA co-localisation) or completely unprotected by RPA (blue arrow). Similarly, unprotected ssDNA[BrdU] foci were also found in APH-treated cells (Supplementary Fig. 10D). These findings provide evidence that the toxin exhausts RPA by oversupply of ssDNA substrate.

We next sought to investigate the order of events leading to RPA exhaustion that underpinned RINGs. It was possible that toxin[WT] nuclease activity (Supplementary Fig. 8A–C) caused an initial burst of DNA damage independently of DNA replication that exhausted RPA. We observed toxin-induced RPA foci in non-replicating G1-arrested cells lacking EdU-labelled DNA (Supplementary Fig. 10E: white arrows). Moreover, the toxin[WT] triggered modest but significant RPA-loading onto chromatin in serum-starved cells suggesting damage independently of DNA replication (Fig. 4c). We hypothesised that DNA breaks generated in G1 together with toxin attack at replication forks would increase the load of ssDNA thereby causing RPA exhaustion in S-phase, which triggers DNA damage manifesting as RINGs. Indeed, while RPA foci were observed in ~7% of intoxicated serum-starved cells, this increased to ~60% in the presence of serum in a toxin[WT]-dependent manner (Fig. 4c, d) showing increased RPA sequestration. Moreover, like RINGs in Fig. 2e, RPA foci accumulated in intoxicated cells that incorporated EdU (Supplementary Fig. 10E: blue arrows) further establishing a direct relationship between DNA replication and toxin-induced ssDNA. We predicted that ssDNA accumulation via aphidicolin treatment would phenocopy the toxin. Sure enough, sustained 24 h incubation with APH induced RINGs (Fig. 4e, f), many of which contained RPA foci (Fig. 4e, g) mirroring the toxin phenotype (Fig. 3c, Supplementary Fig. 10F, G). When APH was removed and cells allowed to recover, neither RPA foci nor RINGs were observed at 24 h establishing that RINGs are triggered by persistent replication stress (Fig. 4e–g). When we added toxin[WT] to APH-treated cells, no additive effect on RPA was observed indicating the pool of RPA was already fully sequestered on ssDNA substrate (Fig. 4g). The data supports a virulence mechanism by which the toxin overwhelms the RPA response by oversupply of ssDNA, which causes RINGs.

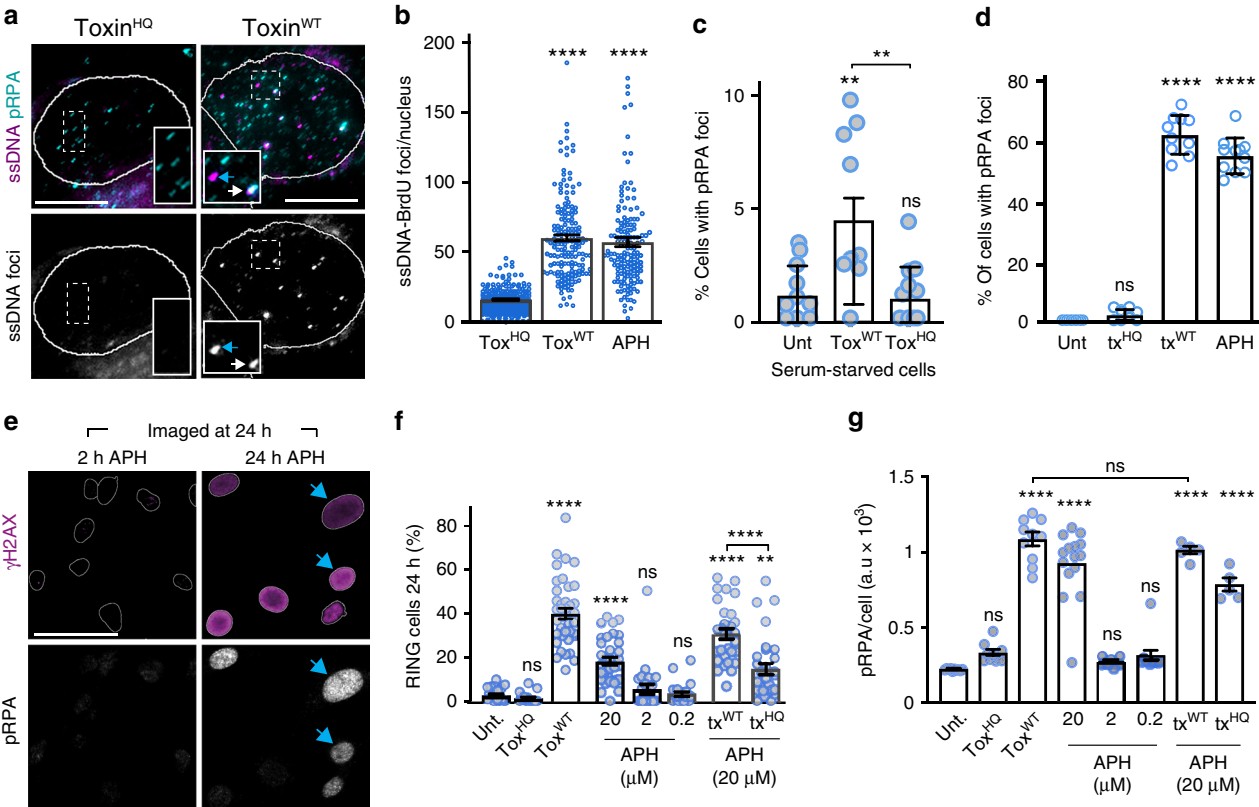

**Fig. 4** Typhoid toxin exhausts RPA by oversupply of ssDNA substrate. **a** Representative image of toxin-induced RPA exhaustion. HT1080 cells intoxicated with toxin^WT or toxin^HQ for 2 h before polymerisation of double-stranded BrdU-labelled DNA from sites of exposed ssDNA template at 24 h (ssDNA, magenta). ssDNA-bound RPA32 pT21 (pRPA, cyan) shown with BrdU shown in grayscale for clarity (ssDNA foci). Insets of toxin^WT magnify ssDNA foci unprotected by pRPA (blue arrow) or partially protected by pRPA indicated by white co-localisation staining (white arrow). Insets of toxin^HQ show RPA-bound ssDNA. Scale bars 10 μm. **b** Proportion of BrdU-labelled ssDNA foci per nucleus in intoxicated HT1080 cells. Aphidicolin (APH) was used as positive control for ssDNA induction and toxin^HQ used as a negative control. Coloured circles indicate means from fields of view. Two biological replicates (160 nuclei per variable). Error bars SEM. **c** Proportion of serum-starved cells with RPA32 pT21 foci (pRPA). Coloured circles indicate means from technical replicates (two biological replicates, 900 nuclei/variable). Error bars SEM. **d** Proportion of cells with RPA32 pT21 foci (pRPA) in presence of serum. Coloured circles indicate means from technical replicates (2 biological replicates, 900 nuclei/variable). Error bars SD. **e** Representative image of RPA32 pT21 (pRPA, grayscale) with γH2AX (magenta) RINGs (blue arrows) in cells treated with APH as indicated. Scale bars 50 μm. **f** Proportion of γH2AX RINGs in APH-treated cells ± toxin^WT as indicated. Coloured circles indicate means from fields of view (two biological replicates, 600 nuclei/variable). Error bars SEM. **g** Mean RPA pT21 (pRPA) foci intensity in APH-treated cells ± toxin^WT as indicated. Coloured circles indicate means from fields of view (two biological replicates, 600 nuclei/variable). Error bars SEM. A.U, arbitrary units. In representative images (**a**, **e**) outlines of DAPI-stained nuclei are shown. Statistical significance (**** = $P < 0.0001$, *** = $P < 0.0002$, ** = $P < 0.0021$, * = $P < 0.0332$, ns = $P > = 0.05$) was calculated relative to toxin^HQ using one-way ANOVA and a Dunnett's multiple comparison test (**b**), or relative to untreated via one-way ANOVA and a Tukey's multiple comparison test (**c**, **d**, **f**, **g**). Source data provided as a Source Data file

**RING-induction by the toxin drives cells into senescence.** We next addressed the biological significance of RINGs. Induction of RPA exhaustion by replication inhibitors causes replication catastrophe and represents a 'point of no return' for cell proliferation, leading to cells acquiring senescent traits[44,46]. Senescence is a phenotype associated with ageing that arises from stress such as persistent DNA damage and underlies multiple age-related pathologies[4]. We thus investigated whether RPA exhaustion by the toxin led to cells acquiring senescent traits[47]: permanent cell-cycle arrest, nuclear distension and augmented lysosomal content (e.g. senescence-associated beta-galactosidase; SA-β-Gal).

We first examined permanent cell-cycle arrest (Fig. 5a). While untreated and toxin^HQ-treated cells proliferated, cells intoxicated with toxin^WT for 2 h failed to re-enter the cell cycle and generate cell colonies over 7-days (top panel). Closer inspection revealed toxin^WT-arrested cells that were substantially enlarged relative to controls cells (arrows in bottom panel), which is consistent with previous reports citing cellular distension[12]. Toxin^WT-treated cells also showed augmented SA-β-Gal (Fig. 5b), which increased from ~40% of cells on day 1 to ~80% by day 6 (Fig. 5c).

We next examined whether the senescence-like phenotype was associated with RING formation. First, we observed toxin-induced SA-β-Gal activity in RING cells directly (blue arrows) but not in cells without DNA damage or nuclear distension (white arrows) (Fig. 5d, Supplementary Fig. 11A). Though serum-starvation is still permissive for SA-β-Gal[48], we still found that the absence of serum, a condition that blocks RING formation (Fig. 2), significantly impaired toxin-induced SA-β-Gal activity further supporting RING-driven senescence. We hypothesised that residual SA-β-Gal in intoxicated serum-starved cells in Fig. 5e was mediated by quiescent cells in G0 and senescent cells arrested in G1 with γH2AX foci (Fig. 2e, Supplementary Fig. 6A). Indeed, in addition to RINGs, single cell analysis also revealed SA-β-Gal activity in cells with γH2AX foci (Supplementary Fig. 11A: white arrow) indicating that the toxin drives a senescence-like phenotype via γH2AX RINGs and foci.

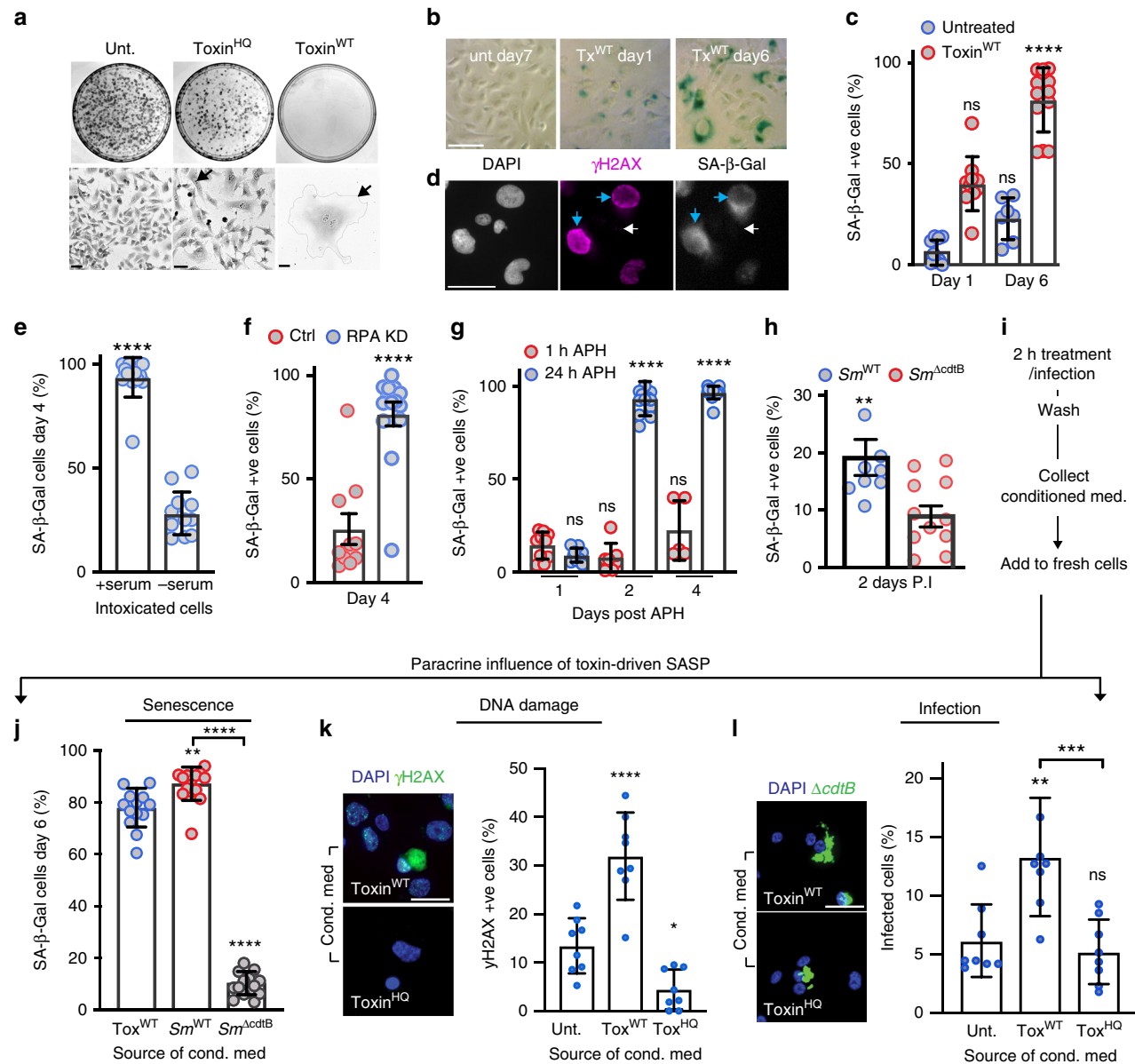

**Fig. 5** RING-induction by the toxin drives senescence and infection. **a** Representative image of toxin-induced cell growth arrest and distension. HT1080s either untreated or treated with toxin[WT] or toxin[HQ] for 2 h before imaging at 7-days. Cell colonies (top panel) and high-magnification images (bottom panel). Arrows mark individual cells. Scale bars 50 μm. **b** Representative image of SA-β-Gal activity in intoxicated cells. Scale bars 50 μm. **c** Proportion of intoxicated cells with SA-β-Gal activity relative to untreated. Analysed 1700 cells/variable, error bars SD. **d** Representative image of toxin-induced SA-β-Gal in γH2AX RING cells at 48 h. Blue arrows indicate senescent RING cells (γH2AX in magenta/SPiDER β-Gal in grayscale) and distended nuclei (DAPI, greyscale). White arrows indicate non-senescent cells. Scale bars 50 μm. Proportion of cells with SA-β-Gal activity following (**e**) intoxication with toxin[WT] ± serum (350 cells/variable, error bars SD), **f** RPA knockdown (900 cells/variable, error bars SEM), **g** APH-treatment (900 cells/variable, error bars SD), and **h** infection with wild-type (Sm[WT]) or toxin-deficient (Sm[ΔcdtB]) S.Javiana (3500 cells/variable, error bars SEM). **i** Experimental pipeline for **j**, **k** and **l** assaying toxin-induced transmissible senescence. Conditioned medium harvested from cells 24 h post-treatment, as indicated, was incubated with fresh cells for 6-days before assaying: **j** SA-β-Gal activity in HT1080s (1240 cells/variable, error bars SD). **k** γH2AX (green) positive HT1080 cell nuclei (DAPI, blue). One biological replicate, 300 cells/variable, error bars SD. Representative γH2AX images shown left. Scale bars 50 μm. **l** Salmonella infection of THP1s. 370 cells/variable, error bars SD. Representative images of THP1s (DAPI, blue) infected with Salmonella[ΔcdtB] (green), left. Scale bars 50 μm. In graphs, coloured circles indicate means from technical replicates (two biological replicates, unless indicated otherwise). Statistical significance (**** = P < 0.0001, *** = P < 0.0002, ** = P < 0.0021, * = P < 0.0332, ns = P > = 0.05) was calculated relative to control (indicated by lack of p-value) using a one-way ANOVA and a Tukey's multiple comparison test (**c**, **g**, **j**, **k**, **l**), or an unpaired two-sided t test (**e**, **f**, **h**). Source data are provided as a Source Data file

Senescence via RINGs was further established by increased SA-β-Gal in RING-inducing conditions via siRPA (Fig. 5f), APH treatment (Fig. 5g) and infection (Fig. 5h). Taken together, these results show that RING formation is associated with a senescence-like phenotype.

**The toxin induces transmissible senescence driving infection.** We next sought to establish whether the typhoid toxin transmits a senescence-associated secretory phenotype (SASP). SASP is a hallmark of cellular senescence characterised by the secretion of proteins into the extracellular milieu that induce secondary

senescence and remodel the functions of bystander cells and tissues[4]. To address toxin-induced SASP (henceforth txSASP), we harvested conditioned cell culture medium from intoxicated cells or *Salmonella*-infected cells before adding the medium to fresh cells and phenotyping as depicted in Fig. 5i. Remarkably, conditioned medium originating from toxin-treated (tox^WT) or wild-type infected (Sm^WT) cells transmitted a senescent-like phenotype to freshly seeded cells (Fig. 5j). In contrast, conditioned medium from cells treated with the toxin null strain had no effect (Fig. 5j; Sm^ΔcdtB). Moreover, successive batches of conditioned medium originating from the Sm^WT-infected cells could also induce txSASP in ~70% of cells (Supplementary Fig. 11B–D). This provides additional evidence of a transmissible senescence-like phenotype whilst also controlling for any toxin secreted by intracellular *Salmonella* during the initial infection (depicted in Supplementary Fig. 11B). Consistent with txSASP, conditioned medium from tox^WT- but not tox^HQ-cells caused a significant DDR (Fig. 5k). We also confirmed that phenotypes were not due to contaminating toxin in conditioned media as wash fractions (depicted in Fig. 5i) were incapable of triggering a γH2AX DDR when added to fresh cells (Supplementary Fig. 11E).

The significance of senescence to host-pathogen interactions is not understood. To investigate whether txSASP influences infection, we treated THP1 cells with conditioned medium 6-days before infection and assaying intracellular *Salmonella* at 24 h (Fig. 5l). We found that txSASP promoted infection (tox^WT) relative to cells treated with control conditioned medium (untreated or tox^HQ). TxSASP enhanced *Salmonella* invasion into host cells (Supplementary Fig. 12A, B), which was enhanced by conditioned medium obtained in RING permissive conditions (+serum) (Supplementary Fig. 12C). Interestingly, SASP by RING-inducers APH and siRPA was markedly reduced relative to the toxin (Supplementary Fig. 12D: APH 40%, siRPA 50%; Fig. 5j: toxin, 80%) and no significant increase in invasion was observed (Supplementary Fig. 12E) indicating that the mode of RPA exhaustion elicited by the toxin initiates a robust SASP, which promotes invasion. These results show that the typhoid toxin triggers a non-canonical DDR that elicits a pathogen-associated senescence-like phenotype, which further drives intracellular *Salmonella* infections by inducing transmissible senescence.

## Discussion

γH2AX is a central hub in the DDR that orchestrates DNA repair and cell fate. Here, we report a non-canonical DDR response characterised by signature accumulation of γH2AX at the nuclear periphery—the RING phenotype—generated via ATR at heterochromatin. Non-canonical γH2AX localisation at the nuclear periphery has been observed in apoptosis[32,33]. This contrasts with the RING phenotype that represents a discrete branch of γH2AX signalling marking replication stress and entry into a senescence-like state.

We uncover that pathogens overwhelm the RPA response to DNA damage. RPA knockdown induced RINGs while increasing RPA expression counteracted the toxin as RINGs were inhibited. Thus, RPA represents a safeguard against RINGs. The cell has a finite pool of RPA for stoichiometric interactions with ssDNA[44], and ssDNA could be left unprotected by (i) low levels of RPA itself (i.e. knockdown experiments), or (ii) high levels of ssDNA generated via a toxin. We show that toxin endonuclease activity elevated the amount of unprotected ssDNA, which sequestered RPA. Consistent with toxin manipulation of RPA, related CDTs induce single-strand DNA breaks in vitro[19,20], as well as accumulate ssDNA and RPA foci in cells[21]. Since heterochromatin is refractory to γH2AX production until its decompaction for DNA replication late in S-phase[1,39,40], it is likely that insufficient RPA

late in S-phase induces DNA damage manifesting as γH2AX RINGs at the nuclear periphery. Thus, our data support a model (Fig. 6) whereby the damage *caused* by the toxin in G1 and S-phase is targeted by RPA (Fig. 6d), which is observed as RPA foci. The damage observed in γH2AX RINGs however is different: this damage is a *consequence* of RPA exhaustion in S-phase, insufficient RPA, which is supported by our observations that (i) RPA is absent from RINGs, and (ii) RPA knockdown is sufficient to induce γH2AX RINGs (Fig. 6e). Together, the data implicate the typhoid toxin and CDTs in pathogen-mediated RPA exhaustion, a virulence mechanism transmitting a senescence-like phenotype that enhances infection (Fig. 6f).

Senescence is a characteristic feature of ageing associated with impaired responses to pathogens[49], yet whether pathogenic bacteria exploit senescence is unclear. We propose that the physiological role of the RING is to induce a senescent-like response to RPA exhaustion, which is hijacked by *Salmonella*. RING cells exhibited senescence-like traits including permanent cell-cycle arrest, persistent DNA damage, augmented lysosomal activity, plus cell and nuclear distension, which have been observed in senescent cells in response to diverse stimuli[47]. Moreover, the toxin triggered senescence in cells directly (i.e. infection, intoxication) and indirectly through txSASP that transmitted the senescence-like effect. Intriguingly, aged animals have been found more susceptible to infections by *Salmonella* Typhimurium[50] and methicillin-resistant *Staphylococcus aureus*[51]. Consistent with this, toxin-dependent transmittable senescence increased host cell susceptibility to *Salmonella* invasion. Nevertheless, whether RINGs occurs in vivo remains to be established: *Salmonella* is known to colonise intestinal crypts, bone marrow, liver and spleen[17,52] that are rich in proliferating cells (i.e. RING-permissive), which could be targeted by secreted toxin. Indeed, secreted typhoid toxin has been localised in the liver where the toxin caused chronic *Salmonella* carriage in an animal infection model[14]. Thus, genotoxic *Salmonella* (e.g. S.Typhi, S.Paratyphi, S. Javiana) could deploy the toxin to subvert senescence in local cellular populations to cause disease, e.g. in chronic human infections[52]. This could be extended to diverse pathogens encoding CDTs that can promote senescence-like phenotypes[53,54].

We reveal a non-canonical DDR, the RING phenotype, triggered by bacterial manipulation of the ssDNA-sensor RPA that causes a senescence-like phenotype to drive *Salmonella* infections. The work uncovers a function of the typhoid toxin that could be of significance to globally important infectious diseases.

## Methods

**Plasmids and recombinant toxin purification.** The plasmid peGFP-C1 encoding *pltB*^His *pltA*^Myc and *cdtB*^FLAG (kind gift from Prof.Teresa Frisan, Umeå) was used to amplify epitope-tagged *pltBA* (PltB_Nco1_FWD 5′-GAGCCCCATGGGCTATA TGAGTAAGTATGTACCCTG-3′/PltA_Myc_EcoR1_REV3′-CGGCCGAATTCTT ACAGGTCTTCTTCAGAGATCAG-5′) and *cdtB* (CdtB_Nde1_FWD 5′-GATAT ACATATGAAAAAACCTGTTTTTTTCCTTC-3′/CdtB_FLAG_Xho1_REV 3′-GG CCACTCGAGTTACTTGTCATCGTCGTCCTTGTA-5′) by polymerase chain reaction (PCR) using Phusion Polymerase (NEB). pETDuet-1 encoding toxin^WT was generated by digesting PCR product *pltBA* and plasmid pETDUet-1-MCS1 using Nco1/EcoR1 restriction endonucleases and *cdtB*/pETDUet-1-MCS2 using Nde1/Xho1 before ligation with T4 DNA ligase according to manufacturer instructions (NEB). The point mutation H160Q was introduced into *cdtB* by PCR site-directed mutagenesis to generate toxin^HQ (CdtB_H160Q_F 5′-GTTTTTCTGA CAGCGCAGGCACTGGCTAGTGGAG-3′/CdtB_H160Q_R 3′-CTCCACTAGCC AGTGCCTGCGCTGTCAGAAAAAC-5′). The plasmid pGEM CDT-I ABC encoding the EcCDT-1 of *E. coli* (strain E6468/62 O86: H34) was a kind gift from Prof. Eric Oswald, Toulouse and was used to amplify epitope-tagged *cdtAB* (ecCdtA_Nco1_F 5′-GGGCCCCATGGATAAAAAACTAATTGCATTTTTGT GC-3′/ecCdtB_Myc_EcoR1_R 3′-CGGCCGAATTCTCACAGATCCTCTTCTGA GATGAGTTTTTGTTCTCTTCTTGCTCCTCTTCCAGGAATAAAGC-5′) and *cdtC* (ecCdtC_Nde1_F 5-GATATACATATGAAAACAGTTATAGTACTTTTTG TTTTACTGC-3′/ecCdtC_His_Xho1_R 3′-GGCCACTCGAGTCAGTGGTGGTG GTGGTGGTGGCTCGTTAATGGAGACATTATTGCCGG-5′) before cloning of

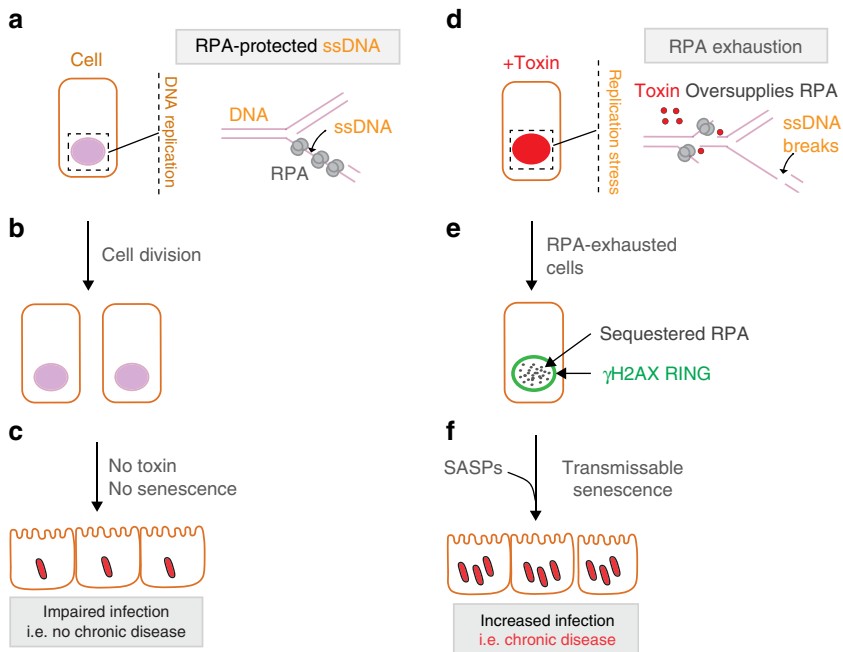

**Fig. 6** Proposed Model. Left: **a** Control cells undergo DNA replication where RPA protects ssDNA from breakage. **b** Cells divide, and **c** no senescence is observed, which impairs *Salmonella* infection. Right: **d** The typhoid toxin causes an initial burst of DNA damage in G1 that combines with damage in S-phase to cause replication stress by oversupply of the RPA substrate ssDNA. **e** RPA sequestration causes DNA damage that manifests as γH2AX RINGs. **f** RING cells enter into a senescence-like state resulting in SASP-induced transmissible senescence that promotes *Salmonella* invasion into host cells. Thus, toxin manipulation of infection niches via senescence may contribute to chronic *Salmonella* infections, which are associated with the typhoid toxin[14,16,57].

EcCDT-1 into pETDuet-1 as typhoid toxin. Recombinant His-tagged toxins were expressed overnight in *E.coli* BL21 DE3 (#69450, Merck Millipore) at 30 °C following addition of 0.1 mM Isopropyl β-D-1-thiogalactopyranoside (Sigma-Aldrich). *E.coli* were harvested by centrifugation and resuspended in lysis buffer containing 20 mM Tris-HCl, pH 8.0, 100 mM NaCl, 1 mM MgCl$_2$ (Sigma-Aldrich) and complete EDTA-free protease inhibitor cocktail (Roche) then lysed by passage through a Cell Disrupter at 30kPSI (Constant systems Ltd). His-tagged toxin was purified from the lysate using Ni-NTA agarose (Qiagen) via affinity chromatography according to manufacturer instructions and stored at −80 °C in lysis buffer supplemented with 20% glycerol.

**Antibodies and immunoblotting.** Antibodies were purchased from Abcam (Salmonella diluted 1:1000, ab35156; RPA70/1:1000, ab79398; RPA32/RPA2/1:1000, ab16850; RPA32pT21/pRPA/1:1000, ab61065; tubulin/1:1000, ab7291/ab52866; H3K9me3/1:1000, ab176916), Cell Signalling Technology (diluted 1:500 CHK1-pS345, 2341; CHK2-pT68/1:500, 2197; ATR-pS428/1:500, 2853; ATM pS1981/1:500, 5883;cleaved PARP1/1:500, 5625; cleaved caspase-3/1:250, 9664; γH2AX/1:1000, 9718), Bethyl laboratories (RPA32 diluted 1:1000, A300-244A; RPA32-pS33/pRPA32/1:1000, A300-246A), Sigma (FLAG M2 diluted 1:1000, F3165), GenScript (Myc diluted 1:1000, A00172-200), Qiagen (His diluted 1:1000, 34660), Millipore (γH2AX diluted 1:1000, 05-636-I), Novusbio (53BP1 diluted 1:1000, NB100-304), BD Bioscience (BrdU diluted 1:200, 347580). Secondary antibodies were diluted 1:10000 and purchased from ThermoFisher Scientific (Alexa 488 donkey anti-mouse IgG, A-21202; Alexa 594 donkey anti-rabbit IgG, A-21207) for immunofluorescence microscopy and LiCor Biosciences (IRDye® 800CW Donkey anti-Mouse IgG, 925-32212; IRDye® 680RD Donkey anti-Rabbit IgG, 926-68073) for immunoblotting. Proteins were separated by 9% Bis-Tris SDS-PAGE gels in MOPS buffer (50 mM MOPS, 50 mM Tris, 0.1% SDS, 20 mM EDTA) and transferred to PVDF transfer stacks using iBlot 2 (Thermo Fisher Scientific). Immunoblotting was performed with Odyssey Blocking buffer (PBS) and IRDye-labelled secondary antibodies according to manufacturer's instructions (LiCor). The unprocessed source data for immunoblots are in the Source Data file.

**Intoxication assay and drug treatments.** Approximately $5 \times 10^4$ mammalian cells were seeded onto glass coverslips (30% confluency) the day before intoxication with 5 ng/ml recombinant typhoid toxin for 2 h at 37 °C. Cells were washed and further incubated, typically 24 h, in complete growth media before phenotypic analysis. Drugs were incubated with cells as indicated in the text or 1 h before intoxication when used in combination with the typhoid toxin and remained present throughout experiments. 20 μM Aphidicolin (A0781, Sigma-Aldrich), 1 μM Killer-TRAIL (ALX-201-073-CO20, Enzo), 100 nM staurosporine (SM97-1 Cambridge Bioscience), 30 μM Z-VAD-OMe-FMK (Selleckchem), 10 μM iATM KU55933, 15 μM iATR NU6072 (Tocris).

**Cell culture and siRNA transfection.** HT1080 (#ATCC® CCL-121), RAW 264.7 (#ATCC® TIB-71™), CACO2 (#ATCC® HTB-37), HAP1 (Horizon Discovery Group) and MEF (#ATCC® SCRC-1008™) cells were cultured (37 °C, 5% CO$_2$) in DMEM (Sigma), HIEC-6 (#ATCC® CRL-3266) cells in OptiMEM 1 Reduced Serum Medium (Gibco), THP1s (#ATCC® TIB-202™) in RPMI1640, and U-2-OS (ATCC identifier HTB-96) cells in McCoy's 5 A Medium (Gibco). Growth media was supplemented with 200 μg/ml⁻¹ streptomycin and 100Uml⁻¹ penicillin (Sigma), 2mM L-Glutamine, and 10% heat-inactivated foetal calf serum (FCS) (Lonza). U-2-OS control cells and U-2-OS cells stably expressing ~two-fold excess of all three RPA subunits via acGFP-RPA3-P2A-RPA1-P2A-RPA2[44] were a kind gift from Prof. Luis Toledo, Copenhagen. THP1 monocytes were differentiated into macrophage-like cells by stimulation with 100 ng/ml Phorbol 12-myristate 13-acetate (PMA) for 3 days then cultured for an additional day without PMA before experiments. Cells were arrested in G$_1$ phase by serum-starvation in serum-free culture medium 24 h before and during experiments. The following were used according to manufacturer instructions: Click-iT EdU Alexa Fluor 647 Imaging Kit and BrdU (Thermo Fisher Scientific), and the Cellular Senescence Detection Kit - SPiDER-ßGal (DOJINDO). ON-Target Plus RPA1 (L-015749-01-0005), ATR (L-003202-00-0005), ATM (L-003201-00-0005) and GAPDH (D-001830-01-05) siRNAs were transfected into cells using DharmaFECT1 (0.25 μl per well of 24-well plate) according to the manufacturer's instructions (Dharmacon) 48 h before experiments.

**Salmonella strains and infection.** S.Javiana (S5-0395) and ΔcdtB (M8-0540)[15,16] were kind gifts from Prof. Martin Weidmann (New York). S.Typhi Ty2 vaccine strain BRD948[55] (kind gift from Prof. Gordon Dougan, Cambridge). The lambda Red recombinase system[56] was used to create S.Typhi ΔcdtB: primers cdtB_FRT_F 5′-ATGAAAAAACCTGTTTTTTTTCCTTCTGACCATGATCATCTTGTGTAGG CTGGAGCTGCTTC-3′/cdtB_FRT_R 3′-TTAACAGCTTCGTGCCAAAAAGGC TACGGGATAATGATCACATATGAATATCCTCCTTAG-5′ were used to amplify a kanamycin-resistance cassette from plasmid pKD4 flanked by FLP (recombinase) recognition target (FRT) sites and ~ 40 nucleotide extensions homologous to cdtB. The construct was transformed into S.Typhi expressing λ red recombinase from the plasmid pKD46 and ΔcdtB transformants cultured overnight at 37 °C on LB agar plates supplemented with 50 μg/ml kanamycin. Gene knockout of cdtB was confirmed by colony PCR using primers flanking cdtB 68_cdtB_up-stream_F 5′-GCACCTTACGCTCAAAG-3′/60_sty1887_3′_R 5′-GAGTTGTTTT ACCAGTC-3′. Kanamycin-resistant ΔcdtB transformants were streaked out onto LB plates without antibiotics and incubated overnight at 42 °C to cure pKD46. For infections, $1 \times 10^5$ of indicated cultured cells were infected with *Salmonella* strains (MOI of 5) for 30 min, washed, and then incubated in growth media supplemented with 10 μg/ml gentamicin to kill extracellular bacteria. When appropriate, conditioned media from *Salmonella*-infected cells was harvested at 48 h and filter-

sterilised before incubation with naïve cultured cells. In microscopy experiments, *Salmonella* were labelled with anti-*Salmonella* antibodies. *Salmonella* invasion was assayed by lysing cells in 0.5% TritonX-100 before culturing *Salmonella* by spotting 10 μl of serially-diluted whole cell lysates on LB agar plates and incubating overnight at 37 °C. The number of *Salmonella* colony forming units (CFUs) were calculated from bacterial colony counts.

**Immunofluorescence microscopy**. Cultured cells on coverslips were fixed by incubation in PBS containing 4% PFA. For immunofluorescence, cells were blocked in PBS 3% BSA 0.2% TritonX-100 and incubations with antibodies performed in PBS 0.2% TritonX-100. DNA was stained with DAPI 6 μg/ml for 3 min. For RPA staining, cultured cells were extracted in PBS 0.1% Tween for 1 min on ice before fixation. Non-differentiated THP1s were fixed and stained in suspension and centrifuged onto microscopy slides for 6 min at 4000 rpm with a cytospin (Shandon Southern Instruments). Coverslips were mounted on slides using VectaShield Antifade Mounting Medium (Vector Laboratories) and sealed with nail varnish. Images were taken on a Nikon Eclipse Ti microscope and collected with a SCMOS camera (Andor Zyla) using NIS-Elements Advanced Research software (Nikon). Quantitative analysis followed high-content immunofluorescence microscopy of cells in 96-well Flat-bottomed μCLEAR plates (Greiner CELLSTAR) with images taken on a Molecular Devices ImageXpress Microscope using MetaXpress 3.1 software. For high resolution immunofluorescence microscopy, cells were seeded on high precision 22 mm × 22 mm cover glasses (Paul Marienfeld GmbH & Co. KG), before imaging on an OMX Deltavision structured illumination microscope or a Zeiss AiryScan confocal microscope (LSM 880) using the Zeiss Zen Software. Microscopy images were processed in ImageJ and figures assembled in Adobe Illustrator. The unprocessed source data for images are in the Source Data file.

**RING tracking**. For automated analysis of DNA damage phenotypes, a custom-made script was developed in MATLAB (http://uk.mathworks.com/products/matlab/) to analyse 20x objective images. First, nuclei were detected using the DAPI channel image: the original image was filtered with a 2-D Gaussian smoothing kernel, which was used to create a binary image with a globally automatically determined threshold. A series of dilation, hole filling and eroding was applied to the binary image with parameters initially manually adjusted by the investigator and then fixed throughout the analysis. The resulting image was used to detect objects, which were filtered by their area (greater than 400px), eccentricity (<0.8), shape regularity (4 × Area$^2$/Perimeter > 0.85), and signal intensity (<2 × Median Intensity of all objects) to exclude all objects that did not represent nuclei. To classify nuclei by γH2AX distribution, the γH2AX signal in each nucleus was binarised with an adaptive threshold, which uses local first-order image statistics around each pixel. This allowed an efficient detection of foci even in the presence of high background. The nuclei with number of foci divided by the nucleus area being greater than 0.0035 were considered as having foci γH2AX distribution. The nuclei with non-foci γH2AX distribution were then further classified into uniform and RING phenotypes. Namely, the profiles of γH2AX intensity from the centre of mass of each object to each pixel at the object boundary were created. These profiles were then resampled to be of the same length, equal to their maximum length, and averaged to create an average radial intensity profile for each nucleus. The nucleus was classified as a RING if the difference between mean intensity in its last quartile and first three quartiles was greater than 10% of overall mean intensity. All above cut-off values were determined empirically by testing images from different experiments and identifying a combination, which was the most efficient in detecting and classifying nuclei. Finally, the correlation between γH2AX and 53BP1 signal was determined by calculating the pairwise correlation coefficient between their intensities at each pixel within individual nuclei.

**Flow cytometry**. Cells were collected with a cell scraper, washed in PBS, and fixed in ice-cold 70% ethanol in PBS on ice for 30 min. Cells were incubated in flow cytometry buffer (PBS, 100 μg/ml RNase, 40 μg/ml propidium iodide, Sigma) for 30 min at 37 °C in the dark to degrade RNA. At least 15,000 cells per sample were subsequently analysed on a FACSCalibur Flow Cytometer (Beckman Coulter). Experiments were done in duplicates and repeated at least three times. Analysis was performed with FlowJo Software.

**Clonogenic survival assay**. Cells were seeded at $2 \times 10^3$ cells per 10 cm dish 24 h before intoxication assays. After 8 days, cells were washed with PBS, fixed for 15 min in 80% ethanol, air-dried and stained with 1% methylene blue for 1 h. Plates were rinsed, dried, and colonies were counted.

**Senescence assays**. To visualise senescent-like cells, a Senescence-Galactosidase Staining Kit (Cell Signaling Technology, 9860) was used according to manufacturer's instruction. The staining was carried out at pH 6.0 in a humidified chamber overnight at 37 °C. Imaging was carried out using an Olympus wide field microscope CK30 10x objective and captured with Dino-Lite adaptor and Dino-Lite software. For SASP experiments, conditioned medium cells was harvested from intoxicated HT1080 cells at 4 days, unless indicated otherwise, and filter-sterilised before incubation with naïve HT1080 cells or non-differentiated THP1s. Cells were incubated for 7 days before assaying DNA damage, infection or SA-β-Galactosidase activity.

**Nuclease assays**. Reactions were prepared on ice and contained Digestion buffer (final concentration 25 mM HEPES pH 7.0, 4 mM MgCl$_2$, 4 mM CaCl$_2$) supplemented with 300 ng pET-Duet1 plasmid DNA and indicated amounts of nuclease. 1U of bovine DNase (Sigma) was used as a positive control. Reactions were performed at 37 °C and stopped by addition of 10x Quench buffer (final concentration 16 mM EDTA, 6% glycerol) before heat inactivation at 75 °C. 100 ng of plasmid DNA was analysed on 0.8% agarose gels buffered in TBE pH8.0 (89 mM Tris, 89 mM Boric acid, 2 mM EDTA). The unprocessed source data for agarose gels are in the Source Data File.

**RPA-protection assay**. Toxin- or aphidicolin-treated cells immobilised on glass coverslips were washed in PBS, permeabilised in PBS 0.1 % Tween for 1 min on ice, then fixed in PBS 4% PFA at room tempertaure. Cells were inverted onto 50 μl DNA polymerase reaction mix containing One-Taq Hot Start polymerase (Bioline, M0481G), 0.4 mM BrdU, 0.2 mM dNTP (Bioline) and incubated for 5 min at 72 °C in a water-bath. Cells were returned to ice and washed with PBS before 20 min permeabilisation with PBS 0.2% TritonX-100. BrdU-labelled DNA double strands were denatured for 30 min with 1 N HCl, washed with PBS, then BrdU labelled by anti-BrdU antibodies before immunofluorescence microscopy.

**Alkaline Comet assay**. Cells seeded in six-well plate (100,000 per well) were treated with typhoid toxin and incubated at 37 °C to permit DNA damage. Cells were harvested using a cell scraper in ice-cold PBS. Cells were combined 1:1 with 1.2% agarose (150 μl) and loaded onto 0.6% agarose (150 μl) immobilised on fully-frosted glass slides then topped with coverglass. Cells were lysed in lysis buffer (10 mM Tris pH10, 100 mM EDTA, 2.5 M NaCl, 1% Tx100) for 1 h, 4 °C. Agarose gel electrophoresis was performed for 25 min at 12 V in electrophoresis buffer (10 mM Tris pH10, 1 mM EDTA, 10 mM NaOH, 1% DMSO). DNA was labelled with SYBR green solution (Invitrogen, S7563). Comets were scored on a microscope with a 10x Plan Fluor objective on Nikon inverted microscope and analysed with software Comet Assay IV (Instem).

**Data analysis and statistics**. Data analysis was performed in Microsoft Excel. Statistical analysis was performed using Prism 7 Graphpad software. Biological replicates are independent experiments. Technical replicates or fields of view indicate analysis of the same experiment. Non-significant is marked as "ns" in figures and legends.

**Reporting summary**. Further information on research design is available in the Nature Research Reporting Summary linked to this article.

## Data availability
The authors declare that data supporting the findings of this study are available within the paper and its Supplementary Information files. A reporting summary for this manuscript is available as a Supplementary Information file. The source data underlying mean values, gel, blot and fluorescence images in Figs. 1–5 and Supplementary Figs. 1–12 are provided as a Source Data file. All new data associated with the paper will be made publicly available on Figshare (https://doi.org/10.15131/shef.data.9449855) and can be identified using the title of the paper as a search term.

## Code availability
The code for RING Tracking is publicly available on github (https://github.com/nbul/Nuclei).

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

## Acknowledgements

We are very grateful to Dr. Catarina Henriques (University of Sheffield) for careful reading of the manuscript and the Department of Biomedical Science (University of Sheffield) for providing a productive and supportive environment for the project. We would like to thank Prof. Gordon Dougan, Prof. Teresa Frisan, Prof. Luis Toledo and Prof. Martin Wiedmann for reagents. Microscopy was performed in the Wolfson Light Microscopy Facility using a Nikon wide-field microscope, an OMX SIM microscope (funded by Medical Research Council SHIMA award MR/K015753/1), and an AiryScan microscope (funded by the Royal Society and the University of Sheffield). The work was funded by a Medical Research Council New Investigator Research Grant to DH (MR/M011771/2) and a British Infection Association Fellowship to AEMI (BIA-2018/FEL/AI). S.E-K is supported by a Wellcome Trust Investigator Award (215485/Z/19/Z) and a Lister Institute of Preventative Medicine Fellowship.

## Author contributions

D.H. supervised the study. A.E.M.I. and D.H. devised the concept. A.E.M.I., D.H., K.N., M.E.G. performed experiments and analysed the data. A.E.M.I., D.H., K.N., S.E.K. designed experiments. S.E.K. designed heterochromatin and DNA comet assay experiments. N.A.B. designed RING Tracking MATLAB algorithm. A.E.M.I, S.E.K., M.E.G., K.N. helped draft and critique the paper for intellectual content. D.H. wrote the paper.

## Additional information

**Competing interests:** The authors declare no competing interests.

