## [Peer Review File · Nature Communications]

Reviewers' comments:

Reviewer #1 (Remarks to the Author):

In this interesting article the authors describe how the relevant pathogen *Salmonella Typhi* interacts with host cells and propose a potential mechanism to explain how the toxins produced by the pathogen might trigger a response that enhances its infective capacity in bystander host cells. The authors show that pathogen-derived toxins with a DNA-nickase activity cause severe DNA-damage in the host cells, which leads to a cellular senescence that can trigger a SASP response in bystander cells, rendering them more susceptible to infection. The authors show that the DNA damage caused by toxins is linked to progression through S-phase and resembles a situation of persistent replication stress that might lead to the exhaustion of RPA due to an excessive generation of ssDNA. This DNA damage arises in the form of gamma-H2AX "RINGS", which the authors identify as a new response that differs from other previously observed rings, such as in conditions of cellular apoptosis.

The results described by the authors are very interesting and show a worthy effort to dissect the molecular mechanism whereby certain pathogens manipulate the cell's physiology to favour infection and their survival. The main advance of this article is a novel kind of toxin-induced DNA damage (RING), and although the authors describe it pretty thoroughly, there are some mechanistic aspects that are still unclear or lack supportive evidence. With some of the following additions I would recommend this article for publication in Nat Comm.

Main points

- * The authors show that the RING response triggers a DDR caused by DNA damage, which suffices to induce a widespread cellular senescence. The authors only show downstream signaling markers of DNA damage, and it would be good to show as well some physical markers of double strand breaks, such as comet assay, PFGE or, if possible, chromosome spreads. In the same line, given the severity of this response, and to further characterise this phenomenon, the authors could provide a relative indication of the degree of DNA damage caused by *S typhi*, comparing their conditions with other DNA damaging agents or insults.
- * The authors show that *S Typhi* causes DNA damage in Sphase, since this doesn't happen in cells arrested in G0/G1. To validate that DNA replication is involved, the authors should show that the DNA damage (RING) appears in cells that incorporate EdU. Ideally, the authors could explore whether there is a direct relationship between DNA replication and *S Typhi*-induced ssDNA, by looking how these two events colocalise in infected cells. Also, the authors describe the response induced by *S Typhi* to be similar to replication stress. To further validate this, the authors should show how DNA replication levels are affected in infected cells.
- * Throughout the paper the authors refer to RPA accumulation as an indicator of ssDNA. However, and at least as a proof of principle, a more direct marker of ssDNA such as native BrDU staining should be used to show ssDNA accumulation in cells infected by *S typhi*, especially in RING+ cells.
- * The authors propose that RINGs appear as a consequence of a RPA exhaustion, since they observe a similar phenotype caused by an RPA knockdown alone, and because SuperRPA cells show a lower level of DNA damage after infection. However some of the data needs clarification. Whether the RPA accumulation that the authors show (like in fig S5E) is the cause or, on the contrary, a consequence of DNA damage is not clear. In fact, the WB in Fig 3 and S5 suggests that RPA accumulation and DNA damage happens simultaneously. The authors could show, as they do for the evolution in the type of DNA damage (from foci to RINGs) how RPA/ssDNA accumulation and the appearance of DNA damage relate in time. Rather than mimicking replication stress (where it is the accumulation of ssDNA which triggers DNA damage after exhausting RPA), it is possible that *S Typhi* causes an initial burst of DNA damage that eventually exhausts RPA, leading to a more severe phenotype (RINGs). The authors should describe the order of these events more thoroughly.

Minor points

The following points are secondary but, if addressed, they could increase the quality of the paper:

- * The results of the RPA kd and S Typhi are confusing (Fig 3C). The authors should explore or argue more in detail why a decreased amount of RPA protein does not synergise with S Typhi infection, if the mechanism behind is RPA exhaustion.
- * To further validate that an extra amount of RPA protects host cells from S Typhi-induced RINGs and Senescence, the authors could explore whether SuperRPA cells can resume proliferation after S Typhi infection.
- * The authors propose that pathogens such as S Typhi enhances its infective capacity by manipulating the host physiology, namely causing a SASP response that enhances the infective capacity in bystander cells (Fig H to K). Following their model, one would expect that cells that enter senescence induced by other means should cause a similar effect in bystander cells. For instance, the authors show that an RPAkd alone causes a similar response (RING), so it would be interesting to explore whether these cells (or other conditions of senescence proposed by the authors) can also trigger a SASP and aid S Typhi infection in bystander cells.

Reviewer #2 (Remarks to the Author):

The manuscript entitled: "Typhoid toxin exhausts the RPA response to DNA replication stress driving senescence and Salmonella infection" by Ibler et al. describes a non-canonical DNA damage response (DRR) in a proportion of the cells exposed to the typhoid toxin or infected with a toxigenic Salmonella for 24h to 48h.

This response is characterized by a specific localization of gH2AX (the phosphorylated form of the histone H2AX, a surrogate marker for DNA damage) at the nuclear periphery, forming a ring-like structure, hence defined as Response-induced by a bacterial genotoxin (RING), which is different and mutually exclusive from the formation of gH2AX foci disperse in the nucleus, as previously observed in cells exposed to DNA damaging agents (1). Interestingly the formation of the RING structures was also mutually exclusive with the formation of 53BP1 foci, another DDR marker (Figure 1). This effect was not specific for the typhoid toxin intoxication, but was also observed in cells infected with a typhoid toxin positive Salmonella or Escherichia coli, expressing another bacterial genotoxin, known as the cytolethal distending toxin (CDT) (Figure 1). Formation of RING was associated with activation of the ATR kinase pathway as assessed by CHK1 and RPA phosphorylation and required transit through the S phase, since this effect was not detected in serum starved cells or in PMA terminally differentiated THP1 macrophages (Figures 2 and 3). siRNA knock out was used to assess the role of RPA, as protein binding single strand DNA (ssDNA), in the RING formation. RPA downregulation was sufficient to cause formation of gH2AX-RING structures in non intoxicated cells. The formation of the gH2AX-RING was not increased by the intoxication when 50nM of RPA specific siRNA was used, suggesting that gH2AX-RINGs were dependent on the exhaustion of RPA and leading unprotected single strand DNA (ssDNA) breaks. Overexpression of RPA reduced the extent of this phenomenon (Figure 3). The authors subsequently used Aphidicolin (APH) and showed that the formation of gH2AX-RINGs was due to persistent replication stress that would exhaust the RPA pool leading to unprotected ssDNA breaks. The APH RING formation was further enhanced by intoxication indicating induction of additional DNA damage over that induced by the APH-mediated stall of the replication fork (Figure 3). In the next set of experiments, the authors demonstrated that intoxication causes a senescent phenotype, and they associated this effect to RING formation and exhaustion of RPA. In addition, they demonstrate that the conditioned medium from senescent cells enhanced Salmonella infection (Figure 4).

The data describes a very interesting phenomenon that surely deserved publication, however there are several issues and questions that need to be addressed, as suggested below.

1. General comment. It is quite difficult to understand what determine the specific localization of gH2AX at the nuclear periphery in certain cells and what is the biological significance of it (see point #7). From the distribution of the phosphorylation of RPA (Figure 3G) one infers that the toxin induces a diffuse DNA damage in all the cell nucleus, however gH2AX accumulate only in the periphery upon RPA exhaustion. Why?

Furthermore, the definition of "Response-induced by a bacterial genotoxin (RING)" is not really appropriate, since RING is formed in other circumstances, as shown by the author themselves, and if, the formation of these structures is dependent on RPA exhaustion, then it should be induced by any agents that cause massive ssDNA breaks, thus a better definition would be gH2AX-RING.

Among the H2AX positive cells how many cells show the RING formation and how many the classical foci distribution? Looking at different figure panels, the % of RING cells seems quite variable, e.g. from approx. 40-50% in Figure 2 to 15% in Figure 3E. The latter shows that there is not dose dependency, which is a bit counterintuitive if the formation of these structures is due on the RPA exhaustion, unless the 5ng/ml toxin concentration is sufficient to exhaust RPA in the intoxicated cells. What happens if the toxin is kept in the medium for all the experimental set up, e.g. 24h, instead of being washed out after 2h, or if the authors uses a higher toxin dose? Would the percentage of RING cells increase due to induction of a high number of ssDNA breaks in more cells? In addition, if a non-efficient protection of ssDNA breaks by RPA is the cause of RING formation, this should be observed also upon inhibition of ATR (but not ATM), which phosphorylates/activates RPA.

2. Figure 1. There is a problem with the 53BP1 staining. 53BP1 should give a uniform nuclear staining in non-treated cells and form discrete foci in cells carrying DNA damage. In addition, there are a lot of cells with 53BP1 foci upon exposure to the mutant toxin. Can the authors comment on this issue? Following this comment, in Figure 1E, the authors should show percentage of positive cells for the markers tested rather than mean fluorescence intensity, which does not provide any specific meaning in the case of the 53BP1 staining, as specified above. Another very puzzling issue is that the gH2AX-RING distribution is mutually exclusive with the formation of the 53BP1 foci. I would have expected formation of 53BP1 foci, at the damage sites, even when there is exhaustion of RPA, why is this not happening?

4. Figure 2. In serum starved cells there should still be induction of DNA damage and activation of the DDR, possibly via ATM which would lead to formation of 53BP1 foci, is this the case? What I'm trying to establish is whether the RING formation is dependent exclusively on the activation of the ATR pathway, and in this case western blot analysis is not the best way to address this issue, since it shows the bulk response, rather than the effect on single cells. As shown in Figure S5A there is clearly an activation of the ATM response upon intoxication, the question is whether the two responses coexist in the RING cells, or these cells will preferentially show a peripheral activation of the ATR pathway.

5. Figures 2B, C, D, E, F and G. The western blot data does not match the immunofluorescence analysis. It is clear that the majority of cells are positive for gH2AX upon serum starvation (Figure 2B), although there are less cells (possibly due to the G0 arrest). This pattern does not correlate with the very faint band observed in western blot (Figure 2C). The same comment applies to Figures 2D/6E and 2F/G. To avoid any problem of interpretation, it would be better to present immunofluorescence data and report them as % of positive cells, rather than the "bulk" western blot analysis. This type of analysis will also allow to assess whether there is a different behavior in pCHK1 in RING versus non-RING cells, as commented in point 4.

6. Figure 3. In Figure 3A, the authors should show the increased levels of RPA in the transfected cells.

Figure 3B, it is puzzling that the transfection per se would induce so much gH2AX. Is this due to stress induction? To investigate whether the gH2AX in control cells is due to stress or induction of

DNA damage, which would create a problem of interpretation, it is possible to assess the levels of the ubiquitinated form of this protein, induced only by DNA damage (2), by western-blot analysis with the gH2AX specific antibody, where the ubiquitinated form appears as a higher molecular weight specie.

Figure 3G: the authors mention in the text that the levels of RPA phosphorylation in APH or APH+Tox treated cells does not change, however, judging from the representative figure, it is clear that the distribution of the staining in the figure is very different in the two conditions: bigger and brighter foci in APH treated cells, smaller and many more less bright foci in the double treatment, which would be consistent with an increased in damaged sites. This difference will not be appreciated by measuring the mean fluorescence intensity, therefore the authors should evaluate at percentage of cells with the two different phenotypes and shows a larger field where more than one cell is present.

7. RING formation and senescent phenotype. The demonstration that the typhoid toxin promotes senescence is expected, since this effect has been previous demonstrated for other genotoxic agents, including bacterial genotoxins such as CDT and colibactin (3, 4). The data presented in the figure do not really support that conclusion that RING induction induces senescence. To prove this the authors should co-localize the presence of senescence marker(s) with RING formation either by using the SenTraGor™ marker or detecting the formation of senescence-associated heterochromatin foci (SAHF) with DAPI by immunofluorescence. This type of co-localization will allow to assess if only RING carrying cells undergo senescence, or whether this happens in all cells showing sign of extensive DNA damage, independently of the gH2AX localization. The interesting observation is that conditioned medium from senescent cells enhanced the cellular susceptibility to Salmonella infection. it would be interesting to see whether the conditioned medium from starved cells or terminally differentiated THP1 macrophages cells has a similar effect, with the due caution to normalize for the number of cells, since starved cells do not grow equally well (as also shown in Figures 2 and S1, see also point 8). Regarding the preparation of the conditioned medium in infected cells, the authors indicate that they wash the cells 2h after infection, and I suppose that they have applied high gentamicin to kill the extracellular bacteria upon bacterial entry, following incubation at low gentamicin. However, the toxin is still released in the supernatant of the infected cells during infection (5, 6), therefore it is likely to be present in the conditioned medium. This step needs to be clarified.

In Figure 4K the authors should show the % of infected cells and the colony forming units (CFUs) recovery: these data will indicate whether the treatment promoted an increase in the number of the infected cells or Salmonella replicates better in the conditioned cells, or both.

8. What is the fate of the starved and intoxicated cells since many of them do not undergo senescence, as shown in Figure 4? In addition, several cells types infected by Salmonella in vivo are terminally differentiated and do not proliferate, this rise the question on whether RING is formed during the course of an in vivo infection, this should be taken in consideration in the Discussion section.

References

1. Rogakou EP, Boon C, Redon C, Bonner WM. 1999. Megabase chromatin domains involved in DNA double-strand breaks in vivo. *J Cell Biol* 146: 905-16
2. Kolas NK, Chapman JR, Nakada S, Ylanko J, Chahwan R, Sweeney FD, Panier S, Mendez M, Wildenhain J, Thomson TM, Pelletier L, Jackson SP, Durocher D. 2007. Orchestration of the DNA-damage response by the RNF8 ubiquitin ligase. *Science* 318: 1637-40
3. Secher T, Samba-Louaka A, Oswald E, Nougayrede JP. 2013. Escherichia coli producing colibactin triggers premature and transmissible senescence in mammalian cells. *PLoS One* 8: e77157
4. Blazkova H, Krejciikova K, Moudry P, Frisan T, Hodny Z, Bartek J. 2010. Bacterial Intoxication Evokes Cellular Senescence with Persistent DNA Damage and Cytokine Signaling. *J Cell Mol Med* 14: 357-67
5. Spano S, Ugalde JE, Galan JE. 2008. Delivery of a Salmonella Typhi exotoxin from a host

intracellular compartment. *Cell Host Microbe* 3: 30-8

6. Guidi R, Levi L, Rouf SF, Puiac S, Rhen M, Frisan T. 2013. *Salmonella enterica* delivers its genotoxin through outer membrane vesicles secreted from infected cells. *Cell Microbiol*

Response to Reviewers

The reviewers were very positive and recommended additional controls to support our conclusions (*indicated with italics*), which have been performed. In summary, the revisions include:

- Fig 1: 4 new panels. They replace 1E that became redundant.
- Fig 2: 4 new panels. They replace 2F, 2G that have moved to S5.
- Fig 3: 4 new panels replacing 3G, 3H, 3I that have moved to Fig 4.
- Fig 4: 4 new panels.
- Fig 5: 1 new panel.
- Fig 6: Improved model.

12 supplementary figures subsuming the former 5 supplementary figures (data in S2 and S4 remain unchanged, former Fig S5 replaced with Fig S6F).

The main text and legends have been revised accordingly to incorporate the new findings and accommodate word limits (**highlighted in blue text**).

The reviewers' points have been addressed point-by-point (indicated by green parentheses in response/main text)

Reviewer 1

- **[R1_1]** R1 says *'it would be good to show as well some physical markers of double strand breaks, such as comet assay'*, which could be compared to *'other DNA damaging agents'* as a control.

The point has been addressed on **page 8** with comet assays that have confirmed typhoid toxin-induced DNA damage, which is now presented **new Supplementary Figure 8B, 8C**. To address the comment on DNA damaging agents, on **page 8** we cite **Fig 1K** showing RING induction by the toxin homologue EcCDT, which is accompanied by a supporting reference where comet assays showed comparable DNA damage caused by EcCDT and the agent etoposide¹.

- **[R1_2A]** R1 *'To validate that DNA replication is involved, the authors should show that the DNA damage (RING) appears in cells that incorporate EdU. [R1_2B] Ideally, the authors could explore whether there is a direct relationship between DNA replication and S Typhi-induced ssDNA, by looking how these two events colocalise in infected cells.'* **[R1_2C]** *'Also, the authors describe the response induced by S Typhi to be similar to replication stress. To further validate this, the authors should show how DNA replication levels are affected in infected cells.'*

[R1_2A] The point has been addressed on **page 6** where **new Fig 2E, 2F** establish RING cells incorporate EdU, thus, validating that DNA replication is involved.

[R1_2B] The point has been addressed with two experiments cited on **page 10** where we used RPA to label toxin-induced ssDNA and EdU to signify DNA replication. First, we found that when cells treated with typhoid toxin initiated DNA replication, RPA foci accumulated (**new Supplementary Figure 10E**) **establishing a direct relationship between ssDNA induction and DNA replication**. Supporting this, RPA-labelled ssDNA was restricted to RING cells (**revised Fig 3C, new Fig 3B**), which were found to be EdU-positive (**new Fig 2E, 2F**). Second, in conditions that were non-permissive for DNA replication (i.e. serum starvation), toxin-induction of RPA foci and RINGs were inhibited (**Fig 2A, new Fig 4C, 4D**). Together, these data establish a relationship between ssDNA induction and DNA replication, **which we state on page 6/7**.

[R1_2C] This point has been addressed on **page 6** where **new Supplementary Figure 6B, 6D** establishes that the toxin inhibits incorporation of BrdU, and thus, showing a reduction in global DNA synthesis.

- **[R1_3]** R1 asks ‘... at least as a proof of principle, a more direct marker of ssDNA such as native BrDU staining should be used to show ssDNA accumulation in cells...’

We have addressed the point on **page 9** where we have developed a proof of principle assay to show accumulation of toxin-induced ssDNA and RPA exhaustion (**new Fig 4A, 4C, Supplementary Figure 10C, 10D**). The basis of the RPA protection assay is depicted in **Supplementary Figure 10C**. In the assay, we exploit the fact that RPA binds and protects ssDNA. We predicted that exposed ssDNA would act as a DNA template for polymerisation of BrDU-labelled DNA by exogenous DNA polymerase, which would be sterically hindered by coats of RPA loaded on ssDNA. Thus, DNA polymerase would preferentially access unprotected tracks of ssDNA in RPA-exhausted cells that could be imaged to assay ssDNA accumulation and RPA exhaustion. Relative to controls, the results show that the toxin^{WT} induces accumulation of ssDNA^{BrDU} foci unprotected by RPA (**new Fig 4A, 4B**), as does RING-inducer aphidicolin (**new Supplementary Figure 10D**), which is known to generate extended tracks of ssDNA². **These findings provide evidence that the toxin exhausts RPA by oversupply of ssDNA substrate, which is stated on page 9 and emphasised in the discussion on page 13.**

- **[R1_4]** R1 asks for clarification: ‘Rather than mimicking replication stress (where it is the accumulation of ssDNA which triggers DNA damage after exhausting RPA), it is possible that *S Typhi* causes an initial burst of DNA damage that eventually exhausts RPA, leading to a more severe phenotype (RINGS). The authors should describe the order of these events more thoroughly.’

We have described the events leading to RPA exhaustion and RING formation more thoroughly on **page 9, page 10** by including new experiments. We now show that the toxin induces an initial burst of ssDNA damage marked by RPA in G1 (serum-starved cells in **new Fig 4C**, and in EdU-negative asynchronous cells indicated by white arrow in **new Supplementary Figure 10E**). The presence of serum increased RPA-loading onto ssDNA increased 10-fold (**new Fig 4C, 4D**). Thus, we state that ssDNA breaks generated in G1 together with toxin-induced breaks at replication forks increase the load of ssDNA thereby causing RPA exhaustion in S phase, which triggers DNA damage manifesting as RINGS. **To make this clear, we have made these points on page 9, 10 and in the legend of our revised model (new Fig 6), which also clarifies the order of events leading to RINGS, also discussed on page 13.**

- **[R1_5]** R1 queries former Fig 3C (now Fig 3G): ‘The authors should explore or argue more in detail why a decreased amount of RPA protein does not synergise with *S Typhi* infection, if the mechanism behind is RPA exhaustion’.

We have addressed this point with a new experiment, which shows synergy between the toxin and siRPA in RING formation (**new Supplementary Figure 8F, cited on page 8**).

- **[R1_6]** R1 suggests ‘To further validate that an extra amount of RPA protects host cells from *S Typhi*-induced RINGS and Senescence, the authors could explore whether SuperRPA cells can resume proliferation after *S Typhi* infection’.

We have addressed the point that an extra amount of RPA protects host cells on **page 9**. The enzymatic activity of typhoid toxin^{WT} was still able to overcome super RPA cells, which failed to resume proliferation following intoxication (**new supplementary Figure 10A, 10B, cited page 9**). **Thus, we state that additional RPA delays RPA exhaustion on page 9.** Nevertheless, we provide further validation that an extra amount of RPA protects host cells from the typhoid toxin in a new immunoblotting experiment where toxin-induction of γ H2AX was markedly reduced in super RPA cells, approximately ~50% (**new Fig 3H**). This mirrored the ~50% reduction in RINGS (**Fig 3I**). This delay in RPA exhaustion was also indicated by the abundance of non-phosphorylated RPA32 (blue arrow) in the population of super RPA cells relative to toxin^{WT}-treated control cells where RPA was predominantly hyper-phosphorylated (red arrow) (**Fig 3H**). **These findings strengthen the**

manuscript by providing more evidence that an extra amount of RPA protects host cells from the typhoid toxin.

- **[R1_7]** R1 *'it would be interesting to explore whether these cells (or other conditions of senescence proposed by the authors) can also trigger a SASP and aid S Typhi infection in bystander cells.'*

We have addressed this point on page 11 with new experiments. Interestingly, SASP by RING-inducers APH and siRPA was markedly reduced relative to the toxin (**new Supplementary Figure 12A**: APH 40%, siRPA 50%; **Fig 5J**: toxin, 80%) and no significant increase in *Salmonella* invasion was observed (**new Supplementary Figure 12B**) indicating that the mode of RPA exhaustion elicited by the toxin initiates a robust SASP, which promotes pathogen invasion (Fig 5L, **new Supplementary Figure 11F, 11G**). **These points are made on page 12.** We predict that differences may also be influenced by the persistence of the toxin inside host cells and damage in G1. This is an area of investigation.

Reviewer 2

- **[R2_1A]** R2 asks *'what determine the specific localization of γ H2AX at the nuclear periphery in certain cells and what is the biological significance of it (see point #7).'*

The biological significance is addressed in point 7 as indicated by the reviewer. The first point on the localisation of γ H2AX has been addressed with **new experiments cited on pages 6, 7 and 8**. Firstly, we reveal that γ H2AX at the nuclear periphery is generated at heterochromatin marked by histone-3 lysine-9 methylation (**new Supplementary Figure 5D, 5E, cited page 6**). Secondly, we show that ATR mediates the specific localisation of γ H2AX RINGs: This was demonstrated using ATR inhibitors that blocked toxin-induced RINGs (**new Fig 2G, 2H, Supplementary Figure 6E, 6F, cited page 7**), and by siATR that blocked siRPA-induced RINGs (**new Supplementary Figure 8E, cited page 8**). **We emphasise that γ H2AX RINGs are generated in an ATR-dependent manner at heterochromatin in a revised discussion on page 13.**

- **[R2_1B]** R2 asks *'From the distribution of the phosphorylation of RPA (Figure 3G) one infers that the toxin induces a diffuse DNA damage in all the cell nucleus, however γ H2AX accumulate only in the periphery upon RPA exhaustion. Why?'*

We address this point with two statements in a revised discussion on page 13, which emphasise that a lack of RPA in exhausted cells causes damage manifesting as RINGs. This view is supported by new experiments and a revised model:

(i) **In the discussion on page 13 we state** *'Our data supports a mechanism whereby the damage caused by the toxin is targeted by RPA, which is observed as RPA foci. The damage observed in γ H2AX RINGs however is different: this damage is a consequence of RPA exhaustion, insufficient RPA, which is supported by our observations that (i) RPA is absent from RINGs, and (ii) RPA knockdown is sufficient to induce γ H2AX RINGs'. **We have revised our model and legend to reflect this view and make it more clear (Fig 6, cited page 13).***

(ii) **In the discussion on page 13** we also make it clear that heterochromatin is refractory to γ H2AX production until heterochromatin is decompacted late in S phase for DNA replication³⁻⁵. We propose that by late S phase heterochromatin accumulates DNA damage due to lack of RPA, which is marked as γ H2AX RINGs. This is supported by new experiments showing that RINGs are generated in S phase (**new Fig 2E, 2F**) at heterochromatin (**new Supplementary Figure 5E, 5F**), and is consistent with RING induction by siRPA.

- **[R2_1C]** R2 indicates *'the definition of "Response-induced by a bacterial genotoxin (RING)" is not really appropriate, since RING is formed in other circumstances.'*

The change has been made on **page 5**. As suggested, we have revised the definition by deleting 'bacterial' to make it broader: ' γ H2AX response induced by a genotoxin (henceforth RING)'.

- **[R2_1D]** *'How many cells show the RING formation and how many the classical foci distribution?'*

This is now quantified in **new Fig 1F**, cited on bottom of **page 4**.

- **[R2_1E]** R2 *'% of RING cells seems quite variable, e.g. from approx. 40-50% in Figure 2 to 15% in Figure 3E. The latter shows that there is not dose dependency'* and R2 asks whether RING formation can be modulated by increasing toxin incubation time or dose.

These points have been addressed on **page 4 and page 5**:

- (i) We predicted that variation in RINGs was a cell line phenomenon (e.g. HT1080s in Fig 2, U2OS in Fig 3). We now show that RING formation varies depending on the cell line, ranging from ~25% to 55% in **new Supplementary Figure S3A, cited page 4**. RINGs were observed in all 7 cell lines tested in the manuscript.
- (ii) We show that RING formation is a dose-dependent effect, which can be modulated by concentration and incubation time (**new Figure 1G, cited page 4**). Interestingly, we find that increasing the toxin dose 10-fold does not increase RING formation. It is probable that G1-arrest in HT1080s limits RINGs (Fig S1E) but this is not the case in U2OS cells where G2 arrest is ~90% (Fig S10A). Thus, to be clear, we state that there is an 'unidentified limiting factor' on the bottom of **page 4/top of page 5**.

- **[R2_1F]** R2 *'In addition, if a non-efficient protection of ssDNA breaks by RPA is the cause of RING formation, this should be observed also upon inhibition of ATR (but not ATM), which phosphorylates/activates RPA.'*

We have addressed this point on **page 7, 8 and 9**. ATR inhibitors blocked RING formation (**new Fig 2G, page 7**) making it an unsuitable approach (see point R2_1A above). Nevertheless, we supply more evidence on RING induction via RPA exhaustion by:

- (i) Developing an 'RPA protection assay' (see **R1_3 above** for detail). **The findings are presented on page 9 in new Fig 4A, 4B, Supplementary Figures S10C, S10D.**
- (ii) Showing synergy between toxin and siRPA in RING formation (**new Supplementary Figure 8F, cited page 8**).

Together, the data provide additional evidence that the toxin exhausts RPA by oversupply of ssDNA substrate.

- **[R2_2A]** R2 *'53BP1 should give a uniform nuclear staining in non-treated cells and form discrete foci in cells carrying DNA damage.'*

We have addressed the point on **page 5** by imaging 53BP1 in various cell lines. As R2 states, we found that DNA damage induced 53BP1 foci in all cell lines (**new Supplementary Figure 5A**). In untreated control cells, 53BP1 was either diffusely localised across the nucleus (HIEC6 cells) or below detection (RAW and MEF, e.g. like HT1080s in Fig 1D). **We make this clear on page 5**. In summary, the results suggest that 53BP1 staining in untreated cells varies in different cell lines.

- **[R2_2B]** R2 on Fig 1D *'there are a lot of cells with 53BP1 foci upon exposure to the mutant toxin.'* *Can the authors comment on this issue?'*

We have commented on this on **page 5** and performed additional analysis in **new Fig 1H**. We have found that relative to untreated controls, the mutant toxin^{HQ} induced a modest but reproducible increase in γ H2AX (**Fig 1D**) and 53BP1 foci (**Fig 1D, 1G**), which was likely due to LPS contamination that binds His-tagged proteins and can activate DDRs⁶⁻⁸. **This**

statement is made on page 5. To avoid confusion, we have replaced the toxin^{HQ} image in Fig 1D to reflect the significant increase in 53BP1 signalling induced by toxin^{WT} (**Fig 1D, 1G**).

- **[R2_2C]** R2 *'in Figure 1E, the authors should show percentage of positive cells for the markers [53BP1, γH2AX] tested rather than mean fluorescence intensity'*.

We have replaced Figure 1E with new experimental data quantifying the percentage of cells with γH2AX phenotypes (**new Fig 1F, cited page 4**) and 53BP1 foci (**new Fig 1H, 1I on page 4, 5**).

- **[R2_2D]** R2 says *'I would have expected formation of 53BP1 foci, at the damage sites, even when there is exhaustion of RPA, why is this not happening?'*

We have addressed this point with new experiments and including a supporting statement with a new citation:

- (i) **On the top of page 6**, we now state *'53BP1 localises to pre-replicative chromatin and is negatively regulated in S phase⁹, which may explain the exclusion of 53BP1 from RINGs (Fig 1D)'*. We now also quantify 53BP1 exclusion from RINGs (**new Fig 1I, page 6**).
- (ii) We support this view with new experiments. Cells with toxin-induced DNA damage foci failed to incorporate nucleotide analogue EdU, demonstrating G1 arrest (**new Fig 2E**), G1-arrest also observed by flow cytometry (**new Supplementary Figure S1E**). G1 arrest has been previously observed for CDTs¹⁰ and the **citation is now included in the introduction on page 3**. Moreover, 53BP1 was maximal in serum-starved cells (**new Supplementary Fig 6D**). This further confirms that γH2AX/53BP1 foci in Fig 1D mark 53BP1-labelled damage on pre-replicative chromatin, which is stated on the **bottom of page 6**.

- **[R2_4]** R2 makes interconnected comments relating to the role of ATR in RING formation. *'...What I'm trying to establish whether the RING formation is dependent exclusively on the activation of the ATR pathway...the question is whether the two responses [ATR/ATM] coexist in the RING cells, or these cells will preferentially show a peripheral activation of the ATR pathway.'*

We have addressed the points on **page 7** in a new section on ATR: We show that ATR is required for RING formation induced by the toxin (**new Fig 2G, 2H, Supplementary Figure 6E, 6F**). Interestingly, γH2AX foci required ATM and signify G1-arrested cells (**new Fig 2G, 2H**, see also γH2AX foci in Edu-negative G1-arrested cells in **2E**). We also show that ATR is required for siRPA-induced RINGs (**new Supplementary Figure 8E, cited page 8**). **We make our conclusions clear on page 7** by stating *'Taken together, these findings show that the typhoid toxin triggers DNA damage foci in G1 through ATM but cell entry into S-phase induces ATR-dependent RING formation in response to replication stress'*. **The role of ATR in RINGs is also emphasised in our revised discussion on page 13.**

- **[R2_5]** R2 on Figure 2: *'The western blot data does not match the immunofluorescence analysis ... it would be better to present immunofluorescence data and report them as % of positive cells, rather than the "bulk" western blot analysis'*

We have addressed the point by **revising Fig 2** and the text on **page 7**. We have removed the 6h image from Fig 2B where the γH2AX signal was stronger than the 24h timepoint in serum starved cells (Fig 2B: serum starved cells +toxin). **The revisions improve the alignment between our γH2AX blots and fluorescence data in figure 2, which makes our results more clear.** Also, we find that γH2AX signal is very robust in RING permissive conditions, which likely relates to a higher concentration of γH2AX present in a cell population with RINGs and foci (+serum) vs foci only (-serum). **We now make this point clear on page 7.** We think this may be significant to other researchers who examine RINGs by immunofluorescence and immunoblotting. Finally, **we now report our γH2AX findings as % of positive cells in Figure 2 (2A, new 2H) and in new Fig 1F, 1G.**

- **[R2_6A]** R2 on superRPA cells in Fig 3 *'the authors should show the increased levels of RPA in the transfected cells'*

We have addressed this point on **page 8 with new Fig 3H**. We had already included the original reference for RPA expression in engineered superRPA cells on **page 8**. In addition, we have now confirmed that superRPA cells express more RPA in a new immunoblotting experiment **Fig 3H** (quantified in **new Supplementary Figure 9**).

- **[R2_6B]** R2 on control transfections in Figure 3B (now Fig 3F) *'it is puzzling that the transfection per se would induce so much γ H2AX. Is this due to stress induction?'*

We have addressed the point on **page 8**. We have repeated knockdown experiments using a lower concentration of DharmaFECT transfection reagent (**stated in the methods on page 16**). The lower concentration enabled RPA knockdown (**revised Fig 3D**) that induced RING formation whilst reducing γ H2AX signal in the siGAPDH control (**revised Fig 3F**). **The revisions show that control transfections do not induce significant γ H2AX, which makes the results on page 8 more clear.**

- **[R2_6C]** R2 asks for clarification regarding the number/size of RPA foci induced by toxin and APH (formerly Fig 3G) *'..the authors should evaluate at percentage of cells with the two different [toxin- and APH-induced RPA] phenotypes and shows a larger field where more than one cell is present.'*

We have addressed the point on **page 8 and 10**.

- (i) We now show a larger field of view of RPA foci in toxin-treated cells (**new Fig 3C**) and APH-treated cells (**new Fig 4E**). The images show a significant number of RPA-positive RING cells, which are quantified in **new Fig 3B, Fig 4D**.
- (ii) We also quantified the size and number of RPA foci, (**new Supplementary Figure 10F, 10G**). No significant difference was observed, which is reflected in the new images with larger fields of view in **Fig 3B, Fig 4D**. **The revisions strengthen our analysis of toxin effects on RPA, which improves the manuscript.**

- **[R2_7A]** In his/her general comments, R2 refers to point 7 with respect to addressing the 'biological significance' of RINGs: *'The data presented in the figure do not really support that conclusion that RING induction induces senescence. To prove this the authors should co-localize the presence of senescence marker(s) with RING formation.... This type of co-localization will allow to assess if only RING carrying cells undergo senescence, or whether this happens in all cells showing sign of extensive DNA damage, independently of the γ H2AX localization'*.

We have addressed the point on **page 10 and 11** with new experiments. To support our conclusions that RING cells are senescent, we performed single cell analysis using the SPiDER- β Gal Cellular Senescence Detection Kit (Dojindo). This enabled us to demonstrate:

- (i) Senescence-associated (SA) β -Galactosidase activity in RING cells directly using fluorescence microscopy (**new Fig 5D, also shown in new Supplementary Figure 11A**).
- (ii) Also that G1-arrested cells with γ H2AX foci contribute to SA- β -Gal activity (e.g. Fig 5E), which is exemplified with SPiDER- β Gal in **new Supplementary Figure 11A**. **Thus, we reveal that the toxin causes senescence via two mechanisms: γ H2AX RINGs and foci, which is stated on page 11.** The mechanisms of senescence by γ H2AX foci are a current focus of investigation.
- (iii) To further support the view that RINGs induce senescence, we already show that the toxin induces cellular distension (**Fig 5A**), a marker of senescence. We now go further by showing that senescent RING cells have distended nuclei (**new Fig 5D**). This supports our evidence

that SA-βGal is induced in RING permissive conditions (i.e. +serum in **Fig 5E**) and by other inducers of RING formation (i.e RPA knockdown in **Fig 5F**, APH treatment in **Fig 5G**).

In summary, the new experiments address the point by providing direct evidence that RING cells induce a senescent-like phenotype. This is emphasised in the **discussion on page 13** to make our results more clear: **'We propose that the physiological role of the RING is to induce a senescent-like response to RPA exhaustion, which is hijacked by *Salmonella*'**.

- **[R2_7B]** R2 asks whether '*conditioned medium from starved cells or terminally differentiated THP1 macrophages cells*' promotes *Salmonella* infection

We have addressed this point on **page 11**. We performed new experiments, which show that *Salmonella* invasion into host cells was enhanced by conditioned medium obtained from RING permissive conditions (+serum) and not serum-starved conditions (**new Supplementary Figure 12C**). **The result strengthens the manuscript by providing additional evidence that toxin-induced SASP in RING permissive conditions promotes infection.**

- **[R2_7C]** R2 asks for an additional control experiment assaying for SASP during infection as '*the toxin is still released in the supernatant of the infected cells during infection*'

We have addressed this point on **page 11**. We performed new experiments to ensure that we are assaying host SASP factors and not toxin secreted during infection by intracellular *Salmonella* (**new Supplementary Figure 11B-D**). We have drawn an experimental workflow for the experiment in **new Supplementary Figure 11B** that has an extended legend to make the results more clear. The resulting SASP phenotypes are shown in **new Supplementary Figure 11C**, which are quantified in **new in 11D**. **The results provide additional evidence and controls for toxin-driven SASP during infection.**

- **[R2_8A]** R2 asks us to clarify '*What is the fate of the starved and intoxicated cells since many of them do not undergo senescence, as shown in Figure 4? [now Fig 5E]*'.

We believe this population in Fig 5E is a mixture of cell-cycle arrested senescent and quiescent cells in serum-free media, a condition permissive for senescence and SA-βGal¹¹ (**citation included on page 11**). This is supported by our flow cytometry experiments showing G0/G1-arrest in serum-starved intoxicated cells (**Supplementary Figure 6A**). New experiments also show that intoxicated cells with γH2AX foci have SA-β-Gal activity (**Supplementary Figure 11A**), which is consistent with induction of γH2AX foci in the absence of serum (**Fig 2**) and residual SA-β-Gal activity in **Fig 5E**. **We make these points on page 10/11 where we clarify the fate of starved and intoxicated cells.**

- **[R2_8B]** R2 says '*several cells types infected by *Salmonella* in vivo are terminally differentiated and do not proliferate, this rise the question on whether RING is formed during the course of an in vivo infection, this should be taken in consideration in the Discussion section*'.

This point is now discussed on page 14.

Bibliography

- 1 Nougayrede, J. P. *et al.* Escherichia coli induces DNA double-strand breaks in eukaryotic cells. *Science* **313**, 848-851, doi:10.1126/science.1127059 (2006).
- 2 Zellweger, R. *et al.* Rad51-mediated replication fork reversal is a global response to genotoxic treatments in human cells. *Journal of Cell Biology* **208**, 563-579, doi:10.1083/jcb.201406099 (2015).
- 3 Gilbert, D. M. Replication timing and transcriptional control: beyond cause and effect. *Curr Opin Cell Biol* **14**, 377-383 (2002).
- 4 Kim, J. A., Kruhlak, M., Dotiwala, F., Nussenzweig, A. & Haber, J. E. Heterochromatin is refractory to gamma-H2AX modification in yeast and mammals. *Journal of Cell Biology* **178**, 209-218, doi:10.1083/jcb.200612031 (2007).
- 5 Polo, S. E. & Jackson, S. P. Dynamics of DNA damage response proteins at DNA breaks: a focus on protein modifications. *Genes Dev* **25**, 409-433, doi:10.1101/gad.2021311 (2011).
- 6 Sewerynek, E. *et al.* Lipopolysaccharide-induced DNA damage is greatly reduced in rats treated with the pineal hormone melatonin. *Mol Cell Endocrinol* **117**, 183-188, doi:10.1016/0303-7207(95)03742-X (1996).
- 7 Mack, L., Brill, B., Delis, N. & Groner, B. Endotoxin depletion of recombinant protein preparations through their preferential binding to histidine tags. *Anal Biochem* **466**, 83-88, doi:10.1016/j.ab.2014.08.020 (2014).
- 8 Cheng, R. *et al.* Gingival fibroblasts resist apoptosis in response to oxidative stress in a model of periodontal diseases. *Cell Death Discov* **1**, doi:ARTN 15046 10.1038/cddiscovery.2015.46 (2015).
- 9 Pellegrino, S., Michelena, J., Teloni, F., Imhof, R. & Altmeyer, M. Replication-Coupled Dilution of H4K20me2 Guides 53BP1 to Pre-replicative Chromatin. *Cell Rep* **19**, 1819-1831, doi:10.1016/j.celrep.2017.05.016 (2017).
- 10 Frisan, T., Cortes-Bratti, X., Chaves-Olarte, E., Stenerlow, B. & Thelestam, M. The Haemophilus ducreyi cytolethal distending toxin induces DNA double-strand breaks and promotes ATM-dependent activation of RhoA. *Cell Microbiol* **5**, 695-707 (2003).
- 11 Yang, N. C. & Hu, M. L. The limitations and validities of senescence associated-beta-galactosidase activity as an aging marker for human foreskin fibroblast Hs68 cells. *Exp Gerontol* **40**, 813-819, doi:10.1016/j.exger.2005.07.011 (2005).

REVIEWERS' COMMENTS:

Reviewer #1 (Remarks to the Author):

The authors have carried out a great effort to meet the concerns I had with the first version of the manuscript. Key mechanistic points in the paper are now more clear and solid, mainly the relationship between DNA damage and DNA replication, and the role RPA exhaustion as the trigger of the RING phenotype. Particularly, now the authors show clearly how modulating the pool of RPA has an impact on the extent of RINGs and DNA damage in general. All main and secondary points have been thoroughly explored, and the authors have contributed with a good number of new assays and experiments. Also considering how the comments from the other reviewer have been addressed, the manuscript offers now a broad and detailed description of this interesting phenomenon, with elegant data, and not less important, a beautiful display across all figures that will help readers in capturing the essence of the story. Nonetheless, the authors have made a great effort in rewriting and adding clarifications, which makes the text rich and robust. Therefore, I recommend the publication of this article in NatComm with no further revision.

Luis Toledo

Reviewer #2 (Remarks to the Author):

The authors have extensively revised the manuscript according to the suggestions from the first revision.

Few minor clarifications:

1. Abstract, lines 23-24. "The toxin overloads the RPA pathway with ssDNA substrate causing RPA exhaustion and senescence". I think that this sentence can be misinterpreted, and the reader may think that only RPA exhaustion (associated with RINGs formation) can induce senescence, while the authors clearly show that also cells presenting gH2AX foci can become senescent (page 11, lines 361-364).
2. Introduction page 3, lines 73-74. There is something odd in the sentence starting with "Toxigenic establish..."
3. Results, page 8, lines 276-277 and Figure 3H. It is difficult to see the band corresponding to the over-expressed RPA in the Western-blot presented. According to the figure legend this band should be marked with a dark blue arrow, which I cannot see in the figure.

Response to Reviewer1

Reviewer 1 has no concerns.

Response to Reviewer 2

The authors have extensively revised the manuscript according to the suggestions from the first revision.

Few minor clarifications:

1. *Abstract, lines 23-24. "The toxin overloads the RPA pathway with ssDNA substrate causing RPA exhaustion and senescence". I think that this sentence can be misinterpreted, and the reader may think that only RPA exhaustion (associated with RINGs formation) can induce senescence, while the authors clearly show that also cells presenting γ H2AX foci can become senescent (page 11, lines 361-364).*

To address the point I have edited the abstract (see below) by adding the sentence 'Senescence was also induced by canonical γ H2AX foci revealing distinct mechanisms'. To accommodate this sentence and the 150 word limit, I have deleted 'and not dependent on canonical DNA repair mediators', which is still true for 53BP1 (i.e. independent of 53BP1-mediated DNA repair) but our discovery in the revised manuscript that ATR plays a prominent role in RING formation makes the statement unclear.

2. *Introduction page 3, lines 73-74. There is something odd in the sentence starting with "Toxigenic establish..."*

The sentence has been edited to makes sense (see track changes).

3. *Results, page 8, lines 276-277 and Figure 3H. It is difficult to see the band corresponding to the over-expressed RPA in the Western-blot presented. According to the figure legend this band should be marked with a dark blue arrow, which I cannot see in the figure.*

Reference to a dark blue error was included in the revised manuscript legend by mistake. The error has now been removed from figure legend 3H.